**Partitioning growing season water balance within a forested boreal catchment using**
**sapflux, eddy covariance and a process-based model**
Nataliia Kozii[1], Kersti Haahti[2], Pantana Tor-ngern[3,4], Jinshu Chi[1], Eliza Maher Hasselquist[1],
Hjalmar Laudon[1], Samuli Launiainen[2], Ram Oren[5], Matthias Peichl[1] Jörgen Wallerman[6],
Niles J. Hasselquist[1*]
[1]Department of Forest Ecology and Management, Swedish University of Agricultural
Science, Umeå, 90183, Sweden
[2]Natural Resources Institute Finland (Luke), Latokartanonkaari 9, 00790 Helsinki, Finland
[3] Department of Environmental Science, Faculty of Science, Chulalongkorn University,
Bangkok 10330, Thailand
[4]Environment, Health and Social Data Analytics Research Group, Chulalongkorn University,
Bangkok 10330, Thailand
[5]Nicholas School of the Environment, Duke University, Durham, 27708, North Carolina,
USA
[6]Department of Forest Resource Management, Swedish University of Agricultural Science,
Umeå, 90183, Sweden
*Correspondence to*: Niles Hasselquist (niles.hasselquist@gmail.com)

**Abstract**

Although it is well known that evapotranspiration ($ET$) represent an important water flux at local to global scales, few studies have quantified the magnitude and relative importance of $ET$ and its individual flux components in high latitude forests. In this study, we combined empirical sapflux, throughfall and eddy covariance measurements with estimates from a process-based model to partition the water balance in a northern boreal forested catchment. This study was conducted within the Krycklan Catchment, which has a rich history of hydrological measurements thereby providing us the unique opportunity to compare the absolute and relative magnitude of $ET$ and its flux components to other water balance components. During the growing season, $ET$ represented *ca.* 85 % of the incoming precipitation. Both empirical results and model estimates suggested that tree transpiration ($T$) and evaporation of intercepted water from the tree canopy ($I_C$) represented 43 % and 31 % of $ET$; respectively, and together was equal to *ca.* 70 % of incoming precipitation during the growing season. Understory evapotranspiration ($ETu$) was less important than $T$ and $I_C$ during most of the study period, except for late autumn when $ETu$ was the largest $ET$ flux component. Overall, our study highlights the importance of trees in regulating the water cycle of boreal catchments, implying that forest management impacts on stand structure as well as climate change effects on tree growth are likely to have large cascading effects on the way water moves through these forested landscapes.

**1 Introduction**

In the hydrological cycle, water enters terrestrial ecosystems mainly through precipitation ($P$). This water leaves terrestrial ecosystems either through evapotranspiration ($ET$) back to the atmosphere or as stream runoff ($Q$). At a global scale, $ET$ accounts for *ca.* 60 % of the annual terrestrial $P$ (Oki and Kanae, 2006); yet the relative importance of $ET$ varies

considerably among different biomes, ranging between 55–80 % of incoming $P$ (Peel et al., 2010). Understanding this variation in $ET$ is crucial, as the difference between incoming $P$ and $ET$ represents the available water in terrestrial ecosystems, which in turn has cascading effects on streamflow (Karlsen et al., 2016;Koster and Milly, 1997), groundwater recharge (Githui et al., 2012) and the ecosystem carbon cycle (Wang et al., 2002;Öquist et al., 2014).

Boreal forests cover $ca.$ 12 million $km^2$ of land area and represents the second largest biome behind tropical forests (Bonan, 2008). Given their large size, boreal forests regulate water and energy fluxes over a vast area and thus play an important role in global hydrology and climatology (Bonan, 2008;Baldocchi et al., 2000;Chen et al., 2018). Boreal forests also play an important role in the global carbon cycle (Goodale et al., 2002); sequestering $ca.$ 0.5 petagrams of carbon annually and storing approximately one third of the global terrestrial carbon (Bradshaw and Warkentin, 2015;Pan et al., 2011). However, few studies have partitioned the water balance in boreal forests (Talsma et al., 2018;Peel et al., 2010;Tor-ngern et al., 2018). In the ones that have, $ET$ has been shown to represent 45-85% of incoming $P$ (Peel et al., 2010).

Such large variation in $ET$ across and within biomes may, in part be explained by the fact that $ET$ represents two fundamentally different water flux pathways in terrestrial ecosystems: (1) transpiration ($T$) through stomata of plants and (2) evaporation from wet surfaces. These two pathways are controlled in different ways and to varying degrees by environmental factors and thus are likely to respond differently to changes in environmental conditions and vegetation dynamics. Specifically, $T$ occurs mainly during the growing season and is thus governed by plant physiological processes, whereas evaporation occurs throughout the year and is strongly controlled by vapor pressure deficit, surface wetness, and aerodynamic conductance (Katul et al., 2012). Thus, quantifying the magnitude and

spatiotemporal variation of $T$ and evaporation separately is crucial to better understanding
how water moves through boreal forest landscapes.
Research investigating the biotic and abiotic controls on $ET$ has a long history, dating
back centuries (Katul et al., 2012;Brutsaert, 1982). However, efforts to separately estimate $T$
and evaporation began in the 1970s (see Kool et al., 2014) and ever since there has been an
increasing number of studies partitioning $ET$ (Stoy et al., 2019;Schlesinger and Jasechko,
2014). There are a number of different approaches and methodology to partition $ET$ into its
individual flux components (Kool et al., 2014), including empirical measurements (Mitchell
et al., 2009;Cavanaugh et al., 2011;Good et al., 2014;Sutanto et al., 2014) as well as a number
of different process based models (Sutanto et al., 2012;Stoy et al., 2019;Launiainen et al.,
2015). Each of these different approaches have their advantages and disadvantages and it has
been shown that the relative contribution of different $ET$ flux components differs depending
on the approach used (Schlesinger and Jasechko, 2014). It has therefore been highlighted that
the use of multiple methods is desirable to more accurately partition $ET$ into it individual flux
components (Stoy et al., 2019).
At a global scale, it was recently estimated that $T$ represents 80 to 90 % of terrestrial
$ET$ (Jasechko et al. 2013). The high estimate of $T/ET$ reported by Jasechko et al. (2013) has
been strongly contested (Coenders-Gerrits et al., 2014), with a more conservative estimate of
$T$ representing *ca.* 60 % of $ET$ being more generally accepted (Wei et al., 2017;Schlesinger
and Jasechko, 2014). Most studies typically partition $ET$ at the stand or plot scale without
considering the broader hydrological cycle (e.g., Cienciala et al., 1997;Grelle et al.,
1997;Wang et al., 2017;Ohta et al., 2001;Iida et al., 2009;Hamada et al., 2004;Maximov et
al., 2008;Warren et al., 2018;Schlesinger and Jasechko, 2014). We are aware of only a few
investigations that have at the catchment scale (Telmer and Veizer, 2000;Sarkkola et al.,
2013), and thus we have little empirical data about how compares to other water fluxes (i.e.,
streamflow) in the terrestrial hydrological cycle.

Transpiration can be further partitioned between canopy trees and understory

vegetation. Few studies have measured understory $T$, yet the ones that have suggest that
understory $T$ represents a small fraction of total $T$ (Kulmala et al., 2011;Palmroth et al., 2014)
but the contribution is strongly dependent on canopy tree structure (Constantin et al.,
1999;Baldocchi et al., 1997;Domec et al., 2012). Similarly, total evaporation can be
partitioned into evaporation of precipitation intercepted by canopy trees ($I_C$) and evaporation
from the forest floor, which includes evaporation from non-stomatal surfaces, bare ground
and open water. At a global scale, $I_C$ represents roughly 20 % of incoming $P$ (Wang et al.,
2007) and in many forested ecosystems $I_C$ represents a substantial portion of total evaporation
(Barbier et al., 2009;Gu et al., 2018). By separating $T$ and evaporation into their different flux
components, it is possible to directly assess the important role trees play in the terrestrial
hydrological cycle.

In this study, we use a combination of empirical data derived from eddy-covariance

and sapflux measurements as well as rain gauges collecting open sky and throughfall
precipitation to partition $ET$ into its individual flux components during a single growing
season in a northern boreal headwater catchment. Additionally, we used a multi-layer, multi-
species soil-vegetation-atmosphere transfer model (APES model based on Launiainen et al.,
2015) as an independent approach to partition $ET$. In doing so, the main objective of this
study was to: *i*) constrain the absolute and relative magnitude of $ET$ flux components by
using both empirical data and model simulations and *ii*) to explore how they vary during the
course of the growing season. This study was conducted within the Krycklan Catchment,
which has a rich history of hydrological measurements (see Laudon et al., 2013;Laudon and
Sponseller, 2018), thereby providing us the unique opportunity to compare different $ET$ flux
components to other water balance components (*i.e.*, streamflow) as well as to directly assess
the important role trees play in the boreal hydrological cycle.

**2. Material and Methods**
*2.1 Study site*
The study was conducted in the 14 ha subcatchment C2 (64.26° N, 19.77° E) within the 68
km$^2$ Krycklan Catchment Study area (Laudon et al., 2013) in northern Sweden (Fig. 1). The
Krycklan Catchment Study area is unique as it is one of the oldest long-term catchment
monitoring sites in northern latitudes with continuous hydrological and meteorological
measurements dating back to the early 1980s (Laudon et al., 2017). The 30-year mean annual
temperature in Krycklan (1986-2015) is 2.1° C; with highest mean monthly temperature in
July and lowest temperature in January (14.6°C and -8.6°C; respectively). Mean annual
precipitation is 619 mm yr$^{-1}$, with the majority *(ca.* 60%) falling in the form of rain. Soils
within the C2 are dominated by glacial till (84%), predominately of stony, sandy texture on
gneiss and granite. There is considerable variation in the thickness of the humus layer, yet the
average is 8 cm (Odin, 1992). The average slope is 6% and the outlet of the C2 subcatchment
is located at 243 m a.s.l.

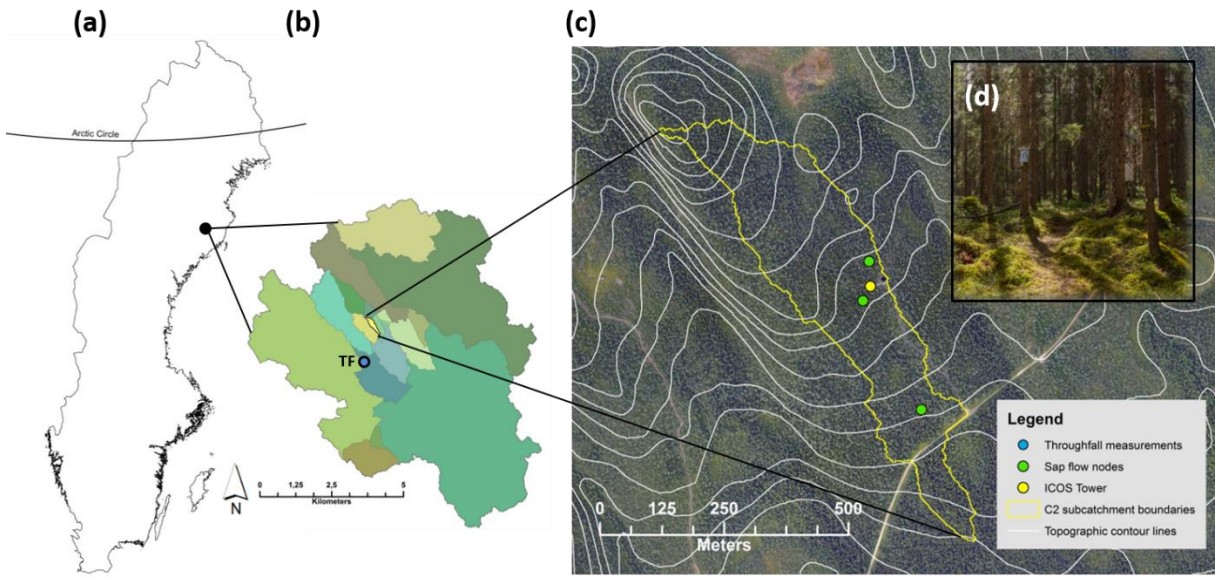


**Figure 1.** Location of the study area in northern Sweden. (a) The outline of Sweden with the location of the Arctic Circle for reference. (b) The boundary of the 68 km$^2$ Krycklan Catchment with various subcatchment in different color; C2 subcatchment in yellow. Throughfall (TF) measurements were made *ca*. 1 km from the C2 subcatchment and are shown on this map (blue circle). (c) High resolution aerial photograph with five-meter contour intervals (white line) and the C2 subcatchment boundary (yellow line). Sap flow measurements were made at three nodes (green circles) and all environmental and eddy-covariance data were taken from the ICOS tower (yellow circle). (d) Picture of the forest stand with understory vegetation that is characteristic of the C2 subcatchment.

The C2 subcatchment is completely covered by an old growth (>100 yr.) mixed forest stand of *Picea abies* (61 %)*, Pinus sylvestris* (34 %), and *Betula* (5 %) (Laudon et al. 2013). The understory consists of a continuous layer of bilberry (*Vaccinium myrtillus*), lingonberry (*Vaccinium vitis idaea*), and mosses (*Pleurozium schreberi* and *Hylocomium splendens*) with no bare ground. Aside from the small (< 0.5 m wide) headwater stream, there is no open water within the C2 subcatchment. Similar forest stands extend to the east and west of the C2 subcatchment boundaries by several hundred meters (Fig. 1c). Within the C2 subcatchment, there is also the Integrated Carbon Observation System (ICOS) Svartberget ecosystem-atmosphere station which provides data on greenhouse gas, water and energy fluxes as well as meteorological, vegetation and soil environmental variables ([www.icos-sweden.se/station_svartberget.html](www.icos-sweden.se/station_svartberget.html)). Our study period was the growing season of 2016. The water balance and *ET* partitioning were restricted to July-October due to measurement availability. The 2016 year was a typical year in terms of precipitation and stream runoff (Fig. S1).

*2.2 Measurements of the water balance components*
We used the hydrological mass balance approach in combination with empirical
measurements of vertical and horizontal water fluxes to quantify the water balance
components within the C2 subcatchment. The mass balance equation is
$$ds/dt = P - ET - Q \qquad (1)$$
where *ds/dt* is change in soil water storage per unit area and *Q* is stream runoff. *ET* was
measured using the eddy covariance technique, and partitioned into components as
$$ET = T + I_C + ET_U \qquad (2)$$
where canopy tree *T* was determined using sap flow sensors and evaporation of intercepted *P*
from the tree canopy ($I_C$) was determined as the difference between open sky precipitation
and water collected on event basis in rain gauges placed below the canopy (see below).
Understory evapotranspiration (*ETu*) was not directly measured in this study, but was instead
calculated as
$$ET_U = ET - I_C - T \qquad (3)$$
Because $I_C$ was estimated on an event basis, our estimate of *ETu* was for the entire growing
season. Daily stream runoff (*Q*) was calculated as daily discharge, obtained from the
Svartberget data portal (https://franklin.vfp.slu.se/), per catchment area. Change in soil water
storage (*ds/dt*), which includes ground water recharge, was calculated as the residual of the
hydrological mass balance (eq. 1).

Environmental data used in this study included open sky precipitation (T200BM

Geonor Inc., New Jersey, USA), air temperature and relative humidity (MP102H Rontronic
AG, Switzerland), wind speed (METEK uSonic3 Class-A, Meteorologische Messtechnik
GmbH, Germany), atmospheric pressure (PTB210 Vaisala Inc., Finland), incoming short and
long-wave radiation (CNR4 Kipp & Zonen B.V., Netherlands), photosynthetic active
radiation (PAR; SQ-110 Apogee Instruments Inc., Utah, USA), as well as soil temperature
and moisture measured at 0.05 m depth (Thermocouple, Type E Campbell Scientific Inc.,
Utah, USA). All environmental data were obtained from the ICOS portal, Svartberget station
(http://www.icos-sweden.se/data.html).
*ET* was obtained from the ICOS-Svartberget eddy covariance (EC) system installed at
32.5 m above the ground. The EC instrumentation consists of a 3D ultrasonic anemometer
(METEK uSonic3 Class-A, Meteorologische Messtechnik GmbH, Germany) for measuring
wind components ($u$, $v$, $w$) and an enclosed infrared gas analyzer (LI-7200, LI-COR
Biosciences, USA) for measuring $CO_2$ and $H_2O$ concentrations. The 10 Hz raw data were
processed in the EddyPro® software (version 6.2.0, LI-COR Biosciences, USA) to obtain the
30-min averaged fluxes. A detailed description of the EC data processing and quality control
can be found in Chi et al. (2019). In brief, the half-hourly *ET* data were corrected for changes
in the storage term which was estimated from concentration profile measurements at several
levels (4, 10, 15, 20, 25 and 30 m) between the forest ground and the measurement height. *ET*
data were then filtered based on the EddyPro quality check flagging policy which includes
tests on steady state and developed turbulent conditions based on Mauder and Foken (2004),
advection effects (Wharton et al., 2009), wind distortion, power failure, and site maintenance
activities. Gaps in the half-hourly *ET* data were filled based on empirical relationships
between *ET* and net radiation using the REddyProcWeb online tool (Wutzler et al., 2018).
Based on the Kljun footprint model (Kljun et al., 2015), the EC footprint (90 %) covers a
measurement area of ~0.5 km$^2$ with a mean upwind fetch of ~400 m surrounding the tower.
The uncertainty in the EC-based *ET* was estimated by the Monte Carlo simulation
(Richardson and Hollinger, 2007).
Evaporation of intercepted *P* from the tree canopy ($I_C$) was determined by subtracting
throughfall (*TF*) from open sky *P*:
$$I_C = P - TF \qquad (4)$$

Previous research within the Krycklan catchment has shown that during the growing season stemflow is negligible in forest stands dominated by *P. sylvestris* and *P. abies* (Venzke, 1990) and consequently omitted in this study. Measurements of *TF* were made 1 km from the study subcatchment (Fig. 1b) by installing 25 rain gauges in a similar mature mixed coniferous forest stand. The design of rain gauges followed WMO (Bidartondo et al., 2001) requirements, which included a stable rim with sharp edge, orifice area of 200 cm$^2$, hydrophobic plastic material and a narrow entrance to the receiving container to prevent evaporation. To test custom made gauges, three of them were installed next to a standardized precipitation collector Geonor T200BM (Geonor Inc., New Jersey, USA) at the Svartberget field station for the entire period and the difference in captured rain was always less than 3%. Measurements of *TF* were made between the beginning of July and the end of October 2016. Water was collected from individual rain gauges immediately after each rain event resulting in event-based $I_C$ estimates (Gash, 1979). Spatial canopy density data acquired from airborne laser scanning (ALS) was used in the FUSION software (McGaughey, 2012) to characterized the canopy structure above each throughfall collector (2 m radius around each collector). We found that the absolute deviation of ALS height measurements from overall median height (ElevMADmedium) showed the highest correlations to $I_C$ and could explain 77% of variation in seasonal $I_C$ (Table S1). $I_C$ within the C2 subcatchment was estimated as a weighted averages of the 25 throughfall collector. The weighting was based on the ElevMADmedium around each throughfall collector and the frequency distribution of this metric within the entire C2 subcatchment. To quantify the uncertainty of event-based $I_C$, we grouped throughfall collectors into five groups based on ElevMADmedium and calculated the standard deviation for each group and event. To eliminate potential difference between open sky *P* within the C2 subcatchment and sampling plot, we estimated the fraction of seasonal interception loss and multiplied that value by cumulative precipitation at the study catchment.

Canopy tree transpiration ($T$) was estimated using sap flux measurements. Within the
EC footprint area, we selected three locations (hereafter referred to as nodes) to measure $T$
(Fig. 1c). Within each node (25 m radius), we selected 20 trees (10 *Pinus sylvestris* and 10
*Picea abies*) that represented the diameter distribution of the entire C2 subcatchment forest
stand. Although *Betula spp.* is also present within the C2 subcatchment, they contribute less
than 5% of the basal area and we therefore focused on the two dominant conifer species
(Laudon et al., 2013).
Sap flux density ($J_S$, g m$^{-2}$sapwood s$^{-1}$) was measured at breast height (1.3 m above
ground) using custom-made heat dissipation-type sap flow sensors (Granier, 1987). Each pair
of sensors consisted of a heated and non-heated probe made from 19-gauge hypodermic
needles with metallic, sensing parts cut into 20 mm length. These sensors were installed on
the selected trees with 10-15 cm spacing between probes and all sensors were covered with
reflective insulation to reduce external temperature influences. To account for azimuthal
(Oren et al., 1999;Lu et al., 2000;James et al., 2002;Tateishi et al., 2008) variation in $J_S$, we
installed sensors in the north, east, south and west sides of the stems in 6 of the selected trees
from all nodes (n = 3 per species). We also installed sensors at four 20 mm interval depths
from the inner bark (i.e., 0-20 mm, 20-40 mm, 40-60 mm and 60-80 mm) in a subset of tree
species to account for radial variation in $J_S$ (Phillips et al., 1996;Ford et al., 2004;Oishi et al.,
2008). Data of temperature difference between the two probes were collected as 30-minute
averages of voltage difference ($\Delta V$, mV) using a data logger (CR1000, Campbell Scientific,
Logan, UT, USA) which was set to record data every 30 s. The collected data were converted
to J$_S$ using the empirical equation  (Granier, 1987)
$$J_S = 118.99 \text{ x } 10^{-6} \times \left(\frac{\Delta V_m - \Delta V}{\Delta V}\right)^{1.231} \tag{5}$$
where $\Delta V_m$ is the maximum voltage difference under zero flow conditions which occur at
night and when vapor pressure deficit is low. We employed the Baseliner program version
4.0 (Oishi et al., 2016) to convert the $\Delta V$ data to $J_S$. This accounts for nocturnal fluxes
resulting from nighttime transpiration and water recharge in stems by selecting the highest
daily $\Delta V$ to represent $\Delta V_m$. The selection criteria for determining $\Delta V_m$ were conditions when
(1) the average, minimum 2-hour vapor pressure deficit is less than 0.02 kPa, thus ensuring
negligible transpiration and (2) the standard deviation of the four highest values is less than
0.5 % of the mean of these values, therefore ensuring that water storage change above the
sensor height is negligible compared to $J_S$.

To determine daily $T$ (mm d$^{-1}$), we first integrated $J_S$ over 24 hours as daily $J_S$ ($J_{SD}$, g

cm$^{-2}$sapwood d$^{-1}$) to avoid issues related to tree water storage and measurement errors (Phillips
and Oren, 1998). Then, we tested $J_{SD}$ variations within sapwood areas in the trees and found
insignificant azimuthal variation (p ≥ 0.23) but significant variation along sapwood depth (p
< 0.001). Accordingly, we performed a scaling based on the radial variation of $J_{SD.}$ First, we
evaluated the relationship between the outermost $J_{SD}$ at 0-20 mm ($J_{SD,0\text{-}20mm}$) sapwood depth
and DBH and found no significant effects of stem size on $J_{SD,0\text{-}20mm}$ in either species (p ≥ 0.1).
Therefore, we averaged $J_{SD,0\text{-}20mm}$ across all sampled trees and used the data for scaling. Next,
we calculated the ratios between $J_{SD}$ at inner sapwood depths (i.e., 20-40 mm, 40-60 mm and
60-80 mm) and $J_{SD,0\text{-}20mm}$ during the study period. Because there was no significant
relationship between the ratios and stem size (p ≥ 0.16), we averaged the ratios across all
trees for each species in each day and used the daily specific ratios between $J_{SD}$ in the inner
sapwood depths and the outermost $J_{SD}$ ($J_{SD,0\text{-}20mm}$) for scaling. Sapwood area ($A_S$, cm$^2$) for
each tree species (*P. sylvestris* and *P. abies*) was estimated from allometric equations derived
from > 20 tree cores taken at breast height for each tree species in 2017. Tree cores were
taken from individual trees representing the full range of stem diameter distribution at the site
and stained with alcohol iodine solution (Eades, 1937) to record the depth of active sapwood
thereby allowing the estimation of $A_S$ of all trees. For scaling, we first estimated weighted
average $J_{SD}$ of each species ($J_{SD,species}$; g cm$^{-2}$ d$^{-1}$) using data from the three nodes by
$$J_{SD,species} = \frac{\sum_{i=1}^{5} J_{SD,i} \times A_{S,i}}{A_{S,all}} \qquad (6)$$

i is the sapwood depth from the inner bark; i.e., 0-20 mm, 20-40 mm, 40-60 mm, 60-80 mm
and >80 mm, $J_{SD,i}$ is the average daily sap flux density for each layer and calculated as the
product of the averaged ratios and $J_{SD,0-20mm}$, $A_{S,i}$ is sapwood area of layer $i$ and $A_{S,all}$ is the
total sapwood area of all trees of the species from all nodes. Then, using this weighted
average $J_{SD}$ by species, the canopy transpiration of the C2 subcatchment ($T$, mm d$^{-1}$) was
estimated using sapwood area index (SAI, m$^2$sapwood m$^{-2}$ground) of each species, which was
derived from data from seven permanent forest inventory plots located within the C2
subcatchment.
$$T = 10 \times (J_{SD,pine} \times SAI_{pine} + J_{SD,spruce} \times SAI_{spruce}) \qquad (7)$$

where 10 is the unit conversion factor. Regarding methodological considerations, the most
common criticism of the heat dissipation method for sap flux measurement, is that it
underestimates the flux (Sun et al., 2012;Steppe et al., 2010). However, according to the
analysis of 54 data from global pine forests in Tor-ngern et al. (2017) estimates from other
sap flux measurement methods showed no particular bias from those with the heat dissipation
one as used in this study. In addition, it has previously been shown that radial variation of sap
flux density and tree size were more important than species in scaling from single-point sap
flux measurements to stand transpiration (2015), both of which were considered in our
analysis. In this study, uncertainty of daily transpiration is represented by standard deviation
of $T$ within the seven permanent forest inventory plots.

*2.3 Modeling ET partitioning and water balance*
We used a slightly modified version of the soil-vegetation-atmosphere transfer model APES
(Launiainen et al., 2015) to partition *ET* and the water balance within the C2 subcatchment
during the studied growing season. APES simulates coupled water, energy, and carbon cycles
in a forest ecosystem consisting of a multi-layer, multi-species tree stand, understory
vegetation, and a bryophyte layer on the forest floor above a multi-layer soil profile. In
APES, the canopy is conceptualized as a layered horizontally homogeneous porous media
characterized by leaf-area density (LAD, $m^2$ leaves $m^{-3}$) distribution. The model solves the
transfer and absorption of shortwave and longwave radiation (Zhao and Qualls, 2005, 2006)
and the transport of scalars (air temperature, $H_2O$, $CO_2$) and momentum among canopy layers
(here n=100). Partitioning of rainfall between interception and throughfall, as well as the
energy balance of wet leaves are also solved for each canopy layer (Watanabe and Mizutani,
1996). The canopy LAD distribution is the superposition of LAD distributions for each plant
type considered (*e.g.*, main tree species and understory vegetation). Each plant type can have
its unique physiological properties (*i.e.*, parameter values) regulating phenology,
photosynthetic capacity and stomatal conductance.

In APES, the coupled leaf gas and energy exchange is calculated separately for sunlit

and shaded leaves of each plant type and canopy layer using well-established photosynthesis–
stomatal conductance theories (Medlyn et al., 2011;Farquhar et al., 1980) and leaf energy
balance (Launiainen et al., 2015). A separate forest floor component describes water, energy
and $CO_2$ dynamics in the bryophyte layer (Kieloaho and Launianen, 2018;Launiainen et al.,
2015). The model thus allows describing the impact of microclimatic gradients along the
canopy, and to partition water fluxes between canopy layers and tree species as well as
between understory *T* and evaporation.
To model the coupled water-energy-carbon cycles, with specific focus on *ET*
partitioning, the vegetation and soil characteristics at C2 subcatchment were assumed to be
horizontally homogenous. The LAD distributions for the main tree species (*Picea abies*,
*Pinus sylvestris*, and *Betula pendula*) were estimated based on stand inventories from seven
forest plots (10 m radius) within the C2 subcatchment. The frequency distributions of
diameter at breast height for each species were converted into needle/leaf biomass and
canopy height  using allometric equations in Marklund (1988) and Näslund (1936)
respectively. The LAD profiles were then derived applying crown-shape models of
Tahvanainen and Forss (2008), and the specific leaf area values reported in Harkonen et al.
(2015). As there are many uncertainties in estimating LAI based on diameter at breast height
alone, the one-sided stand leaf area index ($LAI_{tot}$) was further scaled to match the LAI
estimated from optical measurements done by LAI-2200C Plant Canopy Analyzer.  The
measured $LAI_{Licor} = 2.75$ $m^2$ $m^{-2}$ (Selin, 2019) was corrected for clumping using a correction
factor 1.6–1.9 (Stenberg et al., 1994), resulting in $LAI_{tot}$ between 4.4 and 5.2 $m^2$ $m^{-2}$. The
normalized LAD distributions of each plant type and stand are shown in Fig. S2. In the
simulations, understory $LAI_{under}$ was 0.4–0.8 $m^2$ $m^{-2}$, and the bryophyte layer characterized as
feather moss. Full list of model parameters is provided in the supplementary Tables S2 and
S3.
As forcing variables, the model uses time-averaged (here ½ hourly) meteorological
variables at a reference level above the canopy. These include *P*, downwelling longwave
radiation, direct and diffuse photosynthetically active and near-infrared radiation, wind speed
(or friction velocity), atmospheric pressure, air temperature, and mixing ratios of $H_2O$ and
$CO_2$. We used measured soil moisture and soil temperature at the depth of 0.05 m as lower
boundary conditions for the model. The half-hourly forcing data were obtained from the
Svartberget ICOS station when available, while meteorological measurements from Degerö
ICOS station (at 15 km distance) were used in gap-filling. Precipitation records from Degerö
were corrected to match the daily precipitation measured at another station (at 1 km distance
from C2 center) before using them for gap filling.
We simulated the period from May to October 2016, and included parameter
uncertainty through parameter ranges for $LAI_{tot}$, $LAI_{under}$, maximum carboxylation rate
($V_{cmax}$) at 25°C and interception capacity (see Tables S2 and S3). To assess model
performance, model results were evaluated at ½ hourly time interval against ecosystem fluxes
(net shortwave and longwave radiation, latent heat, sensible heat and gross primary
productivity) observed at the ICOS-Svartberget EC tower (Chi et al., 2019). Performance test
against the simulation results for the center of the parameter space showed a good agreement
between modelled and measured variables (Fig. S3). Net shortwave and longwave radiation
were predicted with good accuracy while sensible heat flux was slightly overestimated and
latent heat flux consequently underestimated. Model results of *ET* components were analyzed
on a daily or rain event-based time interval and compared against corresponding estimates
derived from empirical measurements.

**3   Results**
Meteorological conditions during the 2016 growing season (Fig. 2) were similar to long term
averages. The highest daily mean temperatures were in the middle of July (*ca.* 20 °C)
followed by a gradual decrease to around 0 °C at the end of October. As observed for air
temperature, photosynthetically active radiation (PAR) peaked at the end of July and then
decreased to less than 20 W m$^{-2}$ at the end of October. Daily vapor pressure deficit (VPD)
ranged between 0 and 1.5 kPa, with a notable peak in the middle of July, which also
corresponded to a peak in air temperature. Total precipitation over the study period was 226
mm, with a strong peak in early August and another at the end of September. These rain
events also resulted in peaks in stream runoff (Fig. 2c).

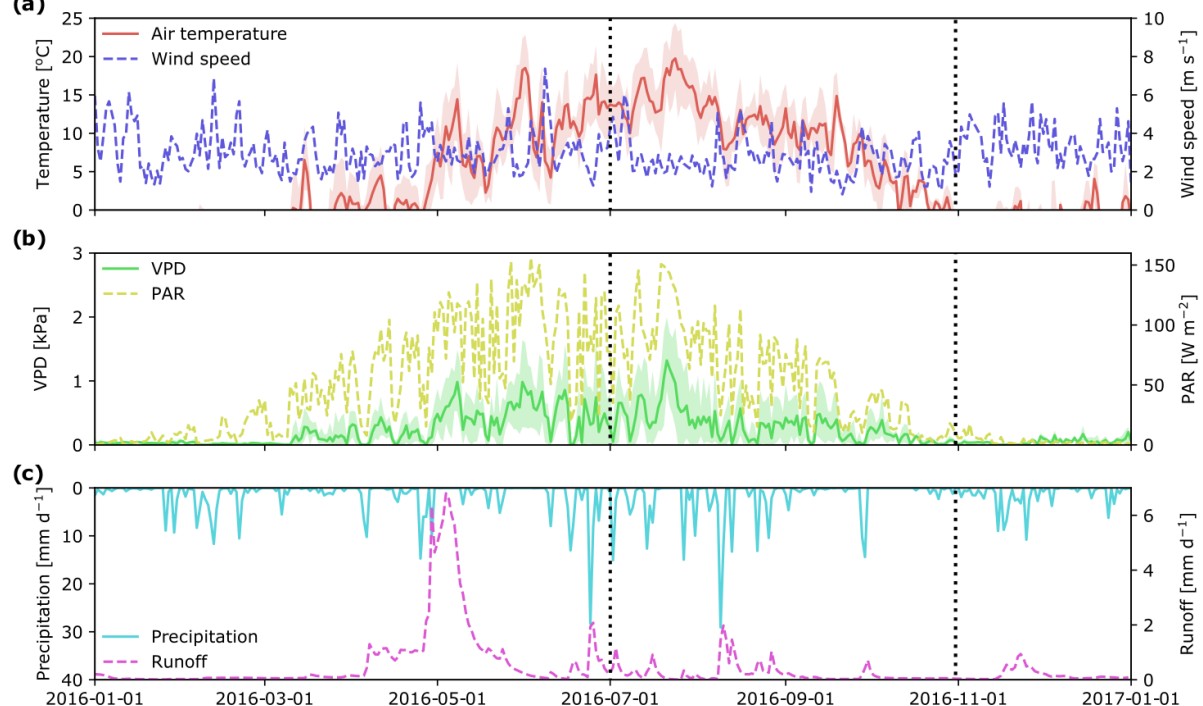


**Figure 2.** Mean daily hydro-meteorological variables at the Krycklan C2 subcatchment
during 2016: air temperature and wind speed (a); vapor pressure deficit, VPD and
photosynthetically active radiation, PAR (b); precipitation and stream runoff (c). Beginning
and end of the study period is marked with vertical dotted lines. Shaded areas for air
temperature and VPD show minimum and maximum values during a day.

*3.1 Daily variability of ET and its components*
Over the study period, daily *ET* varied between 0 and 4 mm d$^{-1}$ depending on the weather
conditions (Fig. 3a). Except for a very short time period following a large rain event on
August 9, *ET* was always higher than *Q*. In general, there was good agreement between
empirical and modeled estimates of *ET* ($R^2$ = 0.79; p < 0.001; Fig. 3a). Yet during a one-
week period in July modeled estimates of $ET$ were 30 % higher than measurement $ET$, which
also corresponded to the time period of high $I_C$ (Fig. 3d).

Canopy transpiration ($T$) was the largest $ET$ flux component, and during 88% of the

study period it alone was higher than $Q$ (Fig. 3b). Maximum daily values of $T$ were reached
during the latter half of July and during this time, the contribution of $T$ to $ET$ was 80%.
During summer months (JJA) and the first half of September, daily $T$ was on average 0.93
mm d$^{-1}$ but later substantially decreased to <0.2 mm d$^{-1}$. Overall, modelled estimates of $T$
were tightly correlated with $T$ based on sap flow measurements ($R^2 = 0.89$; $p < 0.001$),
although the patterns of modelled and measured $T$ diverged during one week in July (Fig.
3b).

Modeled estimates of intercepted $P$ in the tree canopy together with understory

evapotranspiration ($I_C + ETu$) followed a similar pattern to the measured data, which here
was computed as the difference between $ET$ and $T$ (Fig. 3c). Regardless of the approach used,
$I_C + ETu$ had the highest variability throughout the study period (Fig. 3c) mainly because $I_C$
(Fig. 3d) is highly dependent on the frequency of rain events and the effect of other weather
conditions like daily temperature and VPD.

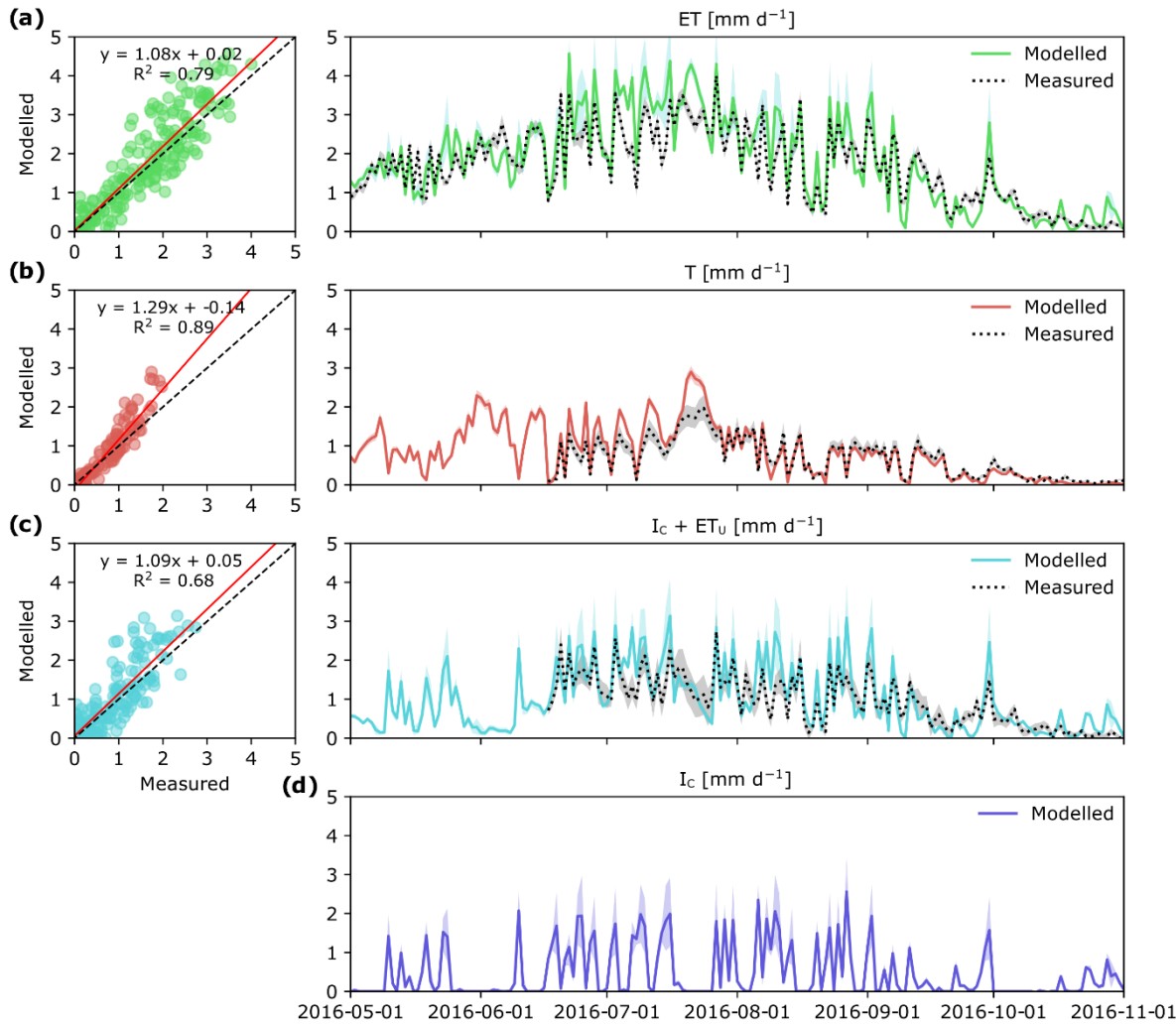


**Figure 3**. Measured and modelled evapotranspiration *ET* (a) and its component fluxes:

canopy transpiration, *T* (b), evaporation of intercepted *P* in the tree canopy and understory

evapotranspiration, $I_C + ETu$ (c) and modeled canopy interception evaporation, $I_C$ (d) in a

boreal forest catchment during the 2016 growing season. Colored shaded areas show

simulation results for whole parameter space and gray shaded areas represent uncertainty in

measurements. Small panels on the left side show correlation between daily modelled and

measured values. Measured $I_C + ETu$ in panel (c) was determined as the difference between

total *ET* and *T*.

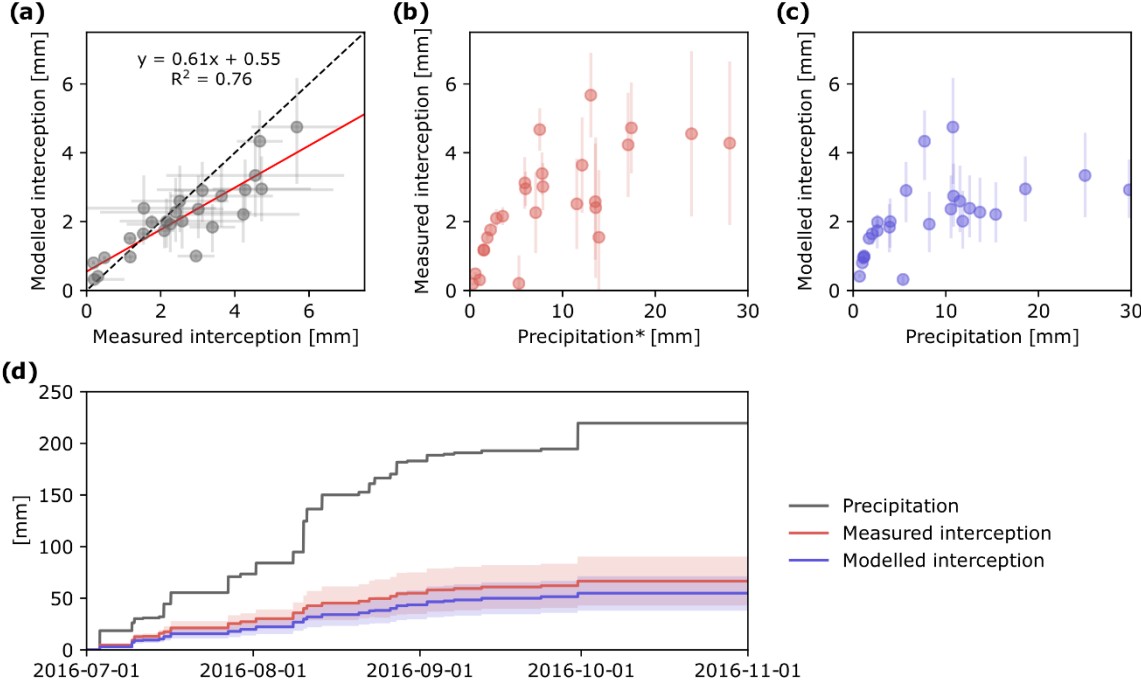

**Figure 4**. Measured and modelled event-based evaporation of *P* in the tree canopy ($I_C$) (a),

relationship between precipitation and measured $I_C$ (b) and modelled $I_C$ (c). Cumulative plot

of precipitation and $I_C$ based on the two different approaches (d). Error bars and shaded areas

show simulation results for whole parameter space and uncertainty range in measurements.

Comparison of measured and modeled event-based $I_C$ showed high correlation

($R^2$=0.76; Fig. 4a). However, modelled $I_C$ values were slightly higher than measured for

small rain events whereas the opposite was true for large rain events (Fig. 4a). Uncertainty of

both measured and modelled $I_C$ increased with the amount of precipitation (Fig. 4b, c).

*3.2 Water balance and ET partitioning*

During the growing season, the C2 subcatchment received 226 mm of *P* and released only 28

mm of water as a stream runoff. Based on EC measurements, *ET* represented 86 % of *P*

during the study period (194 ± 16 mm), which was similar to model estimated that showed

*ET* represented 96 % of *P* (217 ± 18 mm) during the study period (Fig. 5).  Regardless of the
approach used, *T* was the largest ET flux component representing 44 % and 41 % of *ET* based
on empirical measurements and model estimates, respectively.  $I_C$ represented roughly 34 %
(measured) and 28 % (modeled) of *ET*. When combining *T* and $I_C$, trees were responsible for
78 % of *ET* when using empirical data and 69 % based on the model approach. The modeled
*ETu* was slightly higher than that estimated as residual of measured water balance
components (31 % vs. 22 % of *ET*, respectively).

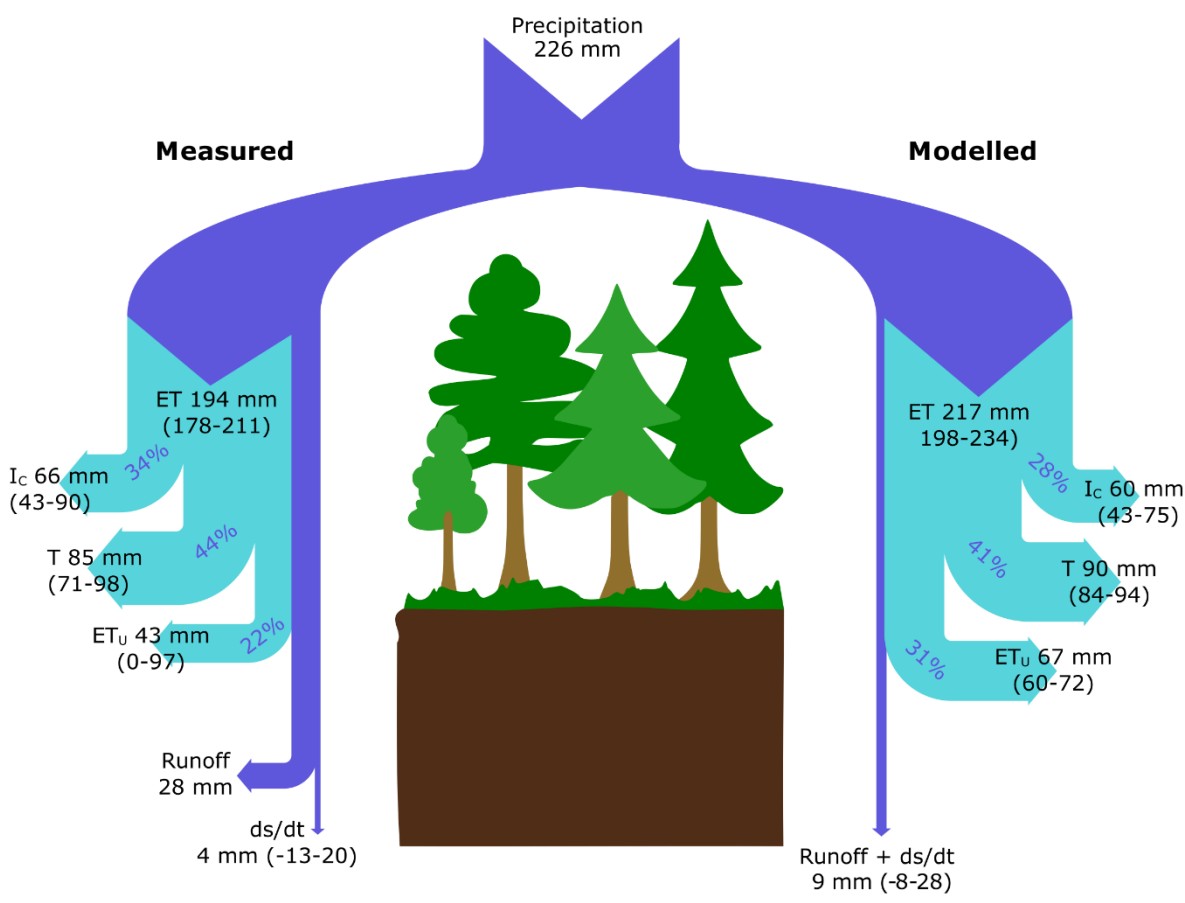


**Figure 5**. Partitioning of water fluxes based on empirical measurements (left side) and model
simulation (right side) in a coniferous boreal catchment during the 2016 growing season
(July-October). Values for each flux are presented as mean absolute values (mm) with upper
and lower boundaries shown in parenthesis. The percentages gives the relative contribution of
*ET* components to total *ET*.


**4. Discussion**

In this study, we used both empirical measurements and a process-based model to partition *ET* into its individual flux components, and assessed how these different fluxes varied during the course of a single growing season in a northern boreal catchment. Both the empirical results and model estimates highlighted the importance of *ET* during the growing season, with *ET* representing *ca.* 85 % of the incoming *P* during the study period. Moreover, the results demonstrated that canopy trees are the main driver of *ET* fluxes during the growing season, as canopy transpiration and evaporation of intercepted rainfall from the canopy jointly represented 69-78 % of *ET* depending on the approach used. Our findings clearly highlight the important role canopy trees play in the boreal hydrological cycle during the growing season, and stresses the need to better understand the effect of trees and their response to forest management practices and a changing climate.

The strong seasonal variation in the relative importance of different water balance components in northern latitude catchments is well known, with stream runoff being the main water flux during snowmelt in spring. Within the Krycklan Catchment, roughly 40 % of annual stream runoff occurs as a response to snowmelt (Ågren et al., 2012), when trees are relatively inactive (Tor-Ngern et al., 2017). In this study, we found that *ET* becomes the dominant water flux after spring flood has ceased, and during the growing season it was seven times greater than stream runoff (Fig. 2c, 3a). In our study, combining *P* with modelled estimates of *ET* and measured stream runoff results in a negative water balance ($P < ET + Q$) during the growing season. This is in agreement with other studies in boreal forests, which have found a negative water balance during the growing season (Wang et al., 2017;Tor-ngern et al., 2018;Sarkkola et al., 2013). Such asynchrony in the relative importance of different water balance components might be even more pronounced in a

future climate when higher air temperatures and less frequent, albeit more intense,
precipitation events can be expected (IPCC, 2018). One future scenario is earlier snow melt
and less snow accumulation during winter as a result of higher air temperatures (Byun et al.,
2019), which would result in earlier peak stream runoff thereby reducing the annual amount
of water available for tree growth during the growing season (Barnett et al., 2005). This, in
turn, could have cascading effects on forest productivity (Barber et al., 2000;Silva et al.,
2010), tree mortality (Peng et al., 2011) and the overall carbon balance in boreal forests (Ma
et al., 2012).

Our results further highlight that $T$ was the largest individual water flux during the

growing season, representing *ca.* 40 % of incoming precipitation. Our cumulative $T$ estimates
during the study period (85-90 mm) were similar in magnitude to previous observations in
other boreal forests (Grelle et al., 1997;Sarkkola et al., 2013). When compared to $ET$, $T$
contributed *ca.* 45 % (Fig. 5), which is also consistent with earlier findings in boreal forest
(Sarkkola et al., 2013;Wang et al., 2017;Ohta et al., 2001), yet lower than the global average
of *ca*. 60 % (Wei et al., 2017;Schlesinger and Jasechko, 2014). However, it is known that the
ratio of $T/ET$ varies considerably among different ecosystems as well as within the same
ecosystems (Evaristo et al., 2015;Wei et al., 2017;Peel et al., 2010). Such variation in $T/ET$
may be the result of differences in study location and duration, its spatial scale, forests stand
structure, climatic conditions as well as the method used (Schlesinger and Jasechko, 2014). It
is important to point out that the two approaches (*i.e.*, empirical measurements and
modelling) gave similar estimates of $T$, both in terms of overall magnitude (Fig. 5) and
seasonal dynamics (Fig. 3b), thereby giving us confidence in the important role canopy tree $T$
plays in the boreal hydrological cycle.

In general, cumulative $I_C$ was the second largest water flux during the study period

(Fig 5). The importance of $I_C$ is not surprising, as $I_C$ has been shown to account for more than
30 % of seasonal $P$ in a wide range of temperate and boreal coniferous forests (Barbier et al.,
2009). In a previous study at the Krycklan catchment, we found that evaporation of
intercepted snow in the tree canopy represents $ca.$ 30 % of winter (November – March)
precipitation (Kozii et al., 2017). Thus, $I_C$ represents the largest $ET$ component when
expressed on an annual time scale as there is negligible $T$ during the winter months (Tor-
Ngern et al., 2017). In our study, $I_C$ was calculated for each rain event and it is important to
point out that the fraction of $P$ lost via $I_C$ ($i.e.$, $I_C/P$) during a single rain event varies in
response to the magnitude and intensity of $P$ (Gash, 1979;Linhoss and Siegert, 2016;Rutter et
al., 1971;Zeng et al., 2000). The highest $I_C/P$ are expected to occur during light rainfall events
in a dry canopy, whereas $I_C/P$ decreases with increasing rain amount and intensity as well as
when water storage capacity in the canopy is reduced by intercepted water from previous
precipitation events.  Thus, projected changes in the amount and frequency of rainfall in
northern latitude ecosystems (IPCC, 2014), could drastically alter $I_C$ and, in turn, strongly
affect the amount of water available to plants, stream runoff and other downstream processes.

Previous studies in boreal forests have shown that understory evapotranspiration

($ETu$) represented 10 – 50 % of $ET$ (Constantin et al., 1999;Iida et al., 2009;Kelliher et al.,
1998;Suzuki et al., 2007;Launiainen et al., 2005;Launiainen, 2010), which is consistent with
our finding in this study. Although $ETu$ was in general less important than $T$ and $I_C$ during the
entire study period, it is worth pointing out that $ETu$ was the largest $ET$ flux component in
late autumn. Using the APES model, we were able to further partition $ETu$ into forest floor
evaporation and understory transpiration. During the study period, model-predicted forest
floor evaporation was 57 mm, representing 85 % of total $ETu$, suggesting that evaporation of
water from the moss layer may play an important role in the boreal hydrological cycle,
especially in late autumn (Bond-Lamberty et al., 2011;Suzuki et al., 2007). However, $ETu$
was the component flux that showed the greatest difference between the two approaches,
which stress the need for additional studies to better quantify *ETu* and its partitioning.

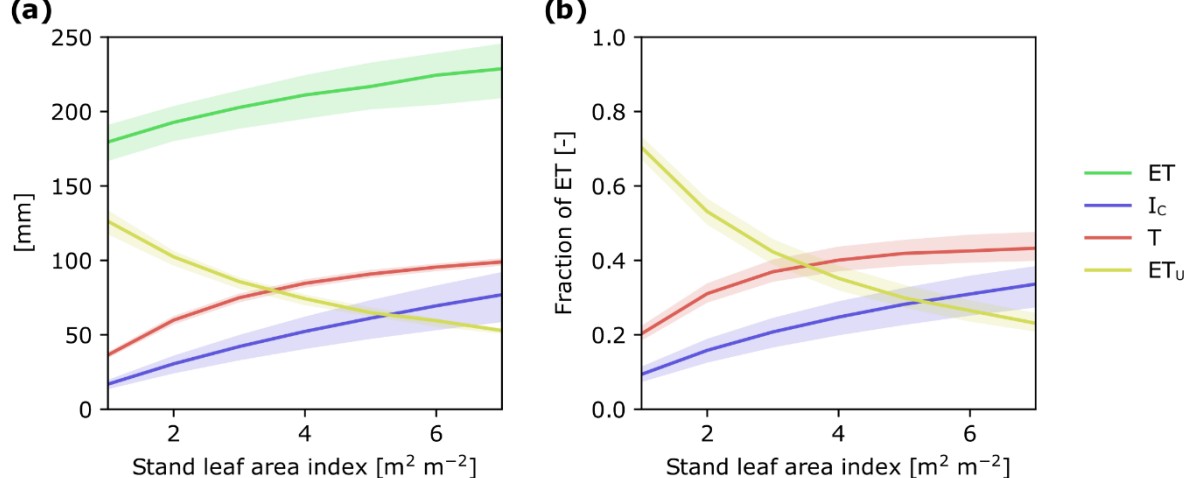


**Figure 6**. Modeled response of *ET* and its flux components to changes in stand LAI: (a) as
cumulative water fluxes and (b) as fraction of *ET* during July-Oct 2016. In simulations,
weather forcing and relative LAD profiles were kept constant and stand LAI varied from 1 to
7 $m^2$ $m^{-2}$. The shaded ranges correspond to model parameter ranges (see Table S2 and S3).

By combing *T* and $I_C$, we are able to show that trees are directly responsible for *ca.* 75
% of *ET* during the growing season. This finding is consistent with other studies in needle-
leaved evergreen forests in boreal and temperate regions that have shown *T* and $I_C$ together
represent 55 to 83 % of *ET* (Gu et al., 2018). Taken together, there is increasing evidence
highlighting the important role trees play in the boreal hydrological cycle. Consequently,
forest management practices that alter forest stand structure could have large cascading
effects on the way water moves through these landscapes (Greiser et al., 2018). For instance,
thinning reduces basal area and LAI of the remaining stand, whereas nitrogen fertilization in
boreal forests promotes greater aboveground carbon allocation leading to an increase in LAI
(Lim et al., 2015) and can also positively affect leaf photosynthetic efficiency and
transpiration (Walker et al., 2014). To assess how forest management practices may affect $ET$
as well as the relative importance of its component fluxes, we ran the APES model with
canopy LAI values ranging from 1 to 7 $m^2\,m^{-2}$. Over this LAI range, $ET$ for the study period
increased by *ca.* 50 mm (Fig. 6a). Fig. 6 also enabled us to identify thresholds in canopy LAI
where the dominant $ET$ component changes. For example, in sparse coniferous stands with
LAI less than 3 $m^2m^{-2}$, understory evapotranspiration appears as the dominant $ET$ component
flux, whereas in forest stands with LAI greater than 3 $m^2\,m^{-2}$ transpiration becomes the
dominant component (Fig. 6b). Understanding how LAI influences $ET$ and its components
fluxes provides an opportunity to assess how different forest management practices may
affect the movement of water in forested landscapes. This, in turn, could assist in the
development of more sustainable management practices (Stenberg et al., 2018;Sarkkola et al.,

2013).


**5. Conclusions**
This study is unique in that it used empirical measurements and a process model approach to
partition the water balance in a northern boreal catchment. In general, the two different
approaches yielded similar results and showed that ET was the main water flux during the
growing season; representing *ca.* 85% of incoming $P$. Moreover, our results highlight the
important role trees play in the boreal hydrological cycle, as canopy $T$ and evaporation of
intercepted P from the tree canopy ($I_C$) together represented *ca. ca.* 75 % of $ET$ during the
growing season. Thus, forest management practices that alter forest stand structure, such as
commercial thinning, continuous cover forestry, and clear cutting, are likely to have large
cascading effects on the way water moves through these forested landscapes. However, it is
important to recognize that this study was limited to a single growing season. It is reasonable
to assume that changes in climatic conditions could also alter the magnitude and relative
importance of different water balance components. Thus, further studies are needed to better
understand how forest management practices and environmental conditions influence *ET* and
its individual flux components in order to identify more sustainable forest management
practices in a changing climate.

**Code and data availability**

Sapflux data is archived in the sapfluxnet data base (https://github.com/sapfluxnet/sapfluxnet-
public/wiki). Data on greenhouse gas, water and energy fluxes as well as meteorological and
environmental data used for model forcing are available through the ICOS portal, Svartberget
station (www.icos-sweden.se/station_svartberget.html). Model source code is available upon
request from Kersti Haahti.

**Author Contributions**

N.K., N.J.H., P.T., R.O., and H.L. worked on the conceptualization of the research goals.
N.K., N.J.H. and P.T. installed, collected and, with the help of R.O., analyzed the sapflux
data; K.H. and S.L. performed the modelling; J.C and M.P. were responsible for processing
the eddy covariance data; E.M.H. and J.W. provided the forest canopy data that was acquired
by airborne laser scanning. N.K. and N.J.H. wrote the paper with contributions from all other
others.

**Acknowledgements**

We thank the research staff at Svartberget and ICOS Sweden for their help in the
establishment and collection of data presented in this manuscript. This work was supported
by grants from the Swedish Research Council (VR, grant number, 2015-04791), the Knut and
Alice Wallenberg Foundation (grant number 2015.0047) and Academy of Finland (decision
number 310203 and 296116). Financial support from the Swedish Research Council and
contributing research institutes to the Swedish Integrated Carbon Observation System (ICOS-
Sweden) Research Infrastructure and the Swedish Infrastructure for Ecosystem Science
(SITES) are also acknowledged. Financial support for Ram Oren was provided by the Erkko
Visiting Professor Programme of the Jane and Aatos Erkko 375th Anniversary Fund through
the University of Helsinki.

**Competing interests**

The authors declare that they have no conflict of interest.

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
