# Peer review of "Partitioning growing season water balance within a forested boreal catchment using"

_Hydrology and Earth System Sciences, 2019_

## Referee Comment (RC1) · Natalie Ceperley (Referee) · 11 Dec 2019

In "Partitioning the forest water balance within a boreal catchment using sapflux, eddy covariance and process-based model", N. Kozii and colleagues, present a comparison of water balance calculations made with measurements vs. a SVAT model in a boreal site in Sweden that has been the location of much intensive monitoring. In this paper, they are primarily bringing together the sapflux and eddy-covariance measurements from a single growing season to produce a calculation of evapotranspiration. In general, I found this work to be important and significant. They present the details of their work well and the comparisons between measurements and between measure-

ments and model are convincing. Nonetheless, I believe that their manuscript could be improved. I found that it often read more like a lab report than an article. I would recommend that in general, they focus on situating their work in current literature / regional and global estimates of ET and improve the story of why their work matters. Additionally, the sentences were often long and cumbersome. I offer a few suggestions (first directly in the pdf file, which became corrupted, and then in the text file), but I believe that with aggressive editing, they could streamline the writing and make it a third shorter, which would improve the readability and allow space for more context.

I strongly recommend that they discuss the boreal ecosystem more in the introduction, that they improve the map to include all measurement points, and that they present the overall water balance equation that they are using in the beginning. They make some rather large assumptions and I believe that presenting the equation early in the paper will make that more clear and help them identify sources of error. I believe their discussion would benefit from a more physical process based explanation centered on the assumptions of their measurements and the model to explain the differences and agreement in the model vs. measurements. Although they do constrain the magnitudes of ET components, they do not really address seasonal variation beyond a description. I would refine this objective to make it clear that the study is limited to a single growing season, or else find a way to expand it to an entire year. What are the implications of the observed seasonal variation? I think the authors have a rather short term view of evaporation and transpiration research and I recommend that they expand their literature search to before the term "ecohydrology" was coined. The title is appropriate. The abstract could be more concise. There are a few points in the text when they mention results that were not shown in full, for example correlations between variables. I think that the supplementary material would benefit from figures and tables showing all relevant information. This would facilitate cross examination of their observations with those in other sites and in other studies. I recommend writing out more of the equations.

[Figure]

I thank them for completing this work and sharing it with the community.

Please also note the supplement to this comment:
https://www.hydrol-earth-syst-sci-discuss.net/hess-2019-541/hess-2019-541-RC1-supplement.zip

---

## Referee Comment (RC2) · Miriam Coenders-Gerrits (Referee) · 16 Jan 2020

The authors studied a boreal forest in Sweden where they executed a detailed study on the partitioning of ET. They measured total ET by means of an EC-system, canopy transpiration by sapflow, and also considered ET from the forest floor. As comparison they applied the APES model to compare they finding. In general, I welcome the study a lot especially, since it are one of the few studies that also consider below canopy evaporation processes in both measuring as modelling. Nonetheless some improvements can be made.

Main concerns:

- L146: the author only consider the growing season. I think it's important to emphasis throughout the paper.

- L176: I am happy to see that below EC-system latent heat is considered. However, equation 1 is only valid once the forest is homogeneous since the footprint of the EC-system and the below canopy latent heat are different. How 'homogeneous is your forest? Please elaborate.

- L199: the TF-sampling was done on 'event-base'. Please elaborate on how this was done. Did you run into your forest after rain ceased? Or did you do daily observations? How did you defined 'an event'?

- L207: How did you tested whether ALS had the highest correlation with seasonal interception loss?

- L220-285: Please have a look at the recent technical note by Larsen et al 2019. Would it be necessary to compensate your sapflow measurements as well? Not doing this could mean an overestimation of your transpiration.

- section 2.3: a better explanation of the modelling principles of APES, would help the reader. For example showing model-scheme.

- section 2: I think it would help to make a schematic picture (a bit like figure 5) of how you define ET and its subcomponents.

- L376-380: be careful with your definitions of transpiration, evaporation and evapo-transpiration. ETu is a combination of forest floor interception, understory transpiration (mosses) and soil evaporation and is thus not only 'evaporation' as said in L378. Also the role of soil evaporation is not explained. Is soil evaporation relevant in your study site? Why/why not.

- Section 4: the discussion and conclusions are merged into one section. I think it would be better to split this. And/or merge the discussion with the results section. But for sure make a separate section for the conclusions only where you are only answering

to the research objective.

Specific (minor) comments:

- L31: redundant to mention "and being roughly 7 times greater than stream runoff". This is the same info as saying ET is 85

- L44: Maybe better to mention the spread in global ET. This is ca 55-80

- L71: after e.g. a comma.

- L128: unit of annual rainfall is mm/year.

- L157-165: variables like P, Q, dS, etc should be in italic.

- L165: I prefer to rename dS into dS/dt, since dS is the storage change per time.

- L172: details => detail.

- Fig S1: the unit of P is mm/y. Furthermore, I would change instead of showing Q/P, showing ET/P. Since this the focus of the paper.

- L337-342: This is a result.

- Fig3c: why showing IL+ETu? Why not only ETu? This would more sense in my view.

- Section 2/fig 3: explain how ETu is 'measured'. It's calculated as ETu=ET-IL-T, right? Please add this equation and elaborate on the fact that ETu is thus not independent of the other measured components.

- Figure 5: I would add the percentages as well. Furthermore, be consistent in the naming of ET and its subcomponents. Would it not be better to use here the abbreviations?

References:

Gleick, Peter H. Water in Crisis: A Guide to the World's Fresh Water Resources. New York: Oxford University Press, 1993.

K. Larsen, E., Palau, J. L., Valiente, J. A., Chirino, E., and Bellot, J.: Technical note on long-term probe misalignment and proposed quality control using the heat pulse method for transpiration estimations, Hydrol. Earth Syst. Sci. Discuss., https://doi.org/10.5194/hess-2019-257, in review, 2019.

---

## Referee Comment (RC3) · Anonymous Referee #3 · 1 Feb 2020

Comments on "Partitioning the forest water balance within a boreal catchment using sapflux, eddy covariance and process-based model" Kozii et al. . .

Title • The word composition of the title is not clear ". . .forest water balance. . ." is it partitioning of water balance in boreal forest, or partitioning forest-water balance? Abstract: • It would be nice to see water balance ways more specific to boreal forests to get a clearer picture how this work is worthy for readers • In line 20, it reads "water is lost"; this is very confusing wording all over the paper. 1) water cannot be lost from a system, 2) I assume this paper deals with water balance, so water "flows" from one state/regime to next, and that is not lost, 3) there could be some

cases where ET can be referred as lost; that is when rainfall is dealt as "gain" • Line 30 change "water loss pathway" to "water balance component" • Line 32 Canopy interception is not part of ET, it should be rather evaporation from canopy • Line 33-34, the numbers do not add up 70, check Introduction: • The study has got no clear definition of hypothesis or purpose of the study • Line 51-52, I don't agree that most studies treat ET as a single water flux pathway • Line 62-63, I think, rather there are dozens of experimental studies for decades • Line 73, what does it mean by "few investigation on water balance at catchment scale"? • The paragraph after line 90 better fits above the previous paragraph • Line 114, what is the state-of-the-art of hydrological measurements at the study site? Give some details of measurements done which of course respective to this study Methods: • Line 147-148, not clear • Line 153-155, not clear • Line 157, what are the environmental data, give the details or examples • Paragraph line 165-175, Too much information. Please classify with instruments, data, how processed, calibrated, purpose – this might help readers to understand • Line 179, what does it mean by "non-stationarity" this word commonly used in statistical description not in instrumentation • Assumptions described in line 188-190 are wrong, re-write (it should be IL = GP-TF-SF) Results and discussion • Are mixed up and not well structured: please take rendering sentences from results to discussion

---

## Author Comment (AC1) · 26 Feb 2020

Dear Natalie Ceperley, thank you for your review and very constructive comments.

Reviewer: Although the word boreal is mentioned, we aren't introduced to the specifics of why it matters or a clear definition, situation on the globe.

*Authors*: We will add a few sentences to the introduction highlighting: (i) the extent of the boreal forest (*i.e.*, one of the largest biomes in the world), (ii) the importance of water fluxes from boreal forest on the global water and energy cycle and (iii) the few studies that have shown large variation in the partitioning of the water balance in these ecosystems (i.e., *ET* represent 45–85 % of incoming *P* in boreal forests).

Reviewer: As someone who has never worked in this part of the world, I have never heard of Krycklan and the term "boreal" does not automatically invoke an idea of what the issues / steaks / interest is regarding the water balance and evapotranspiration. The location is an extremely valuable part of this research and story, please introduce it well.

*Authors*: Similar to the answer above we will added text about the boreal biome, but also better highlight the high latitude location of our study site. This is a geographic area where relatively much less research has been conducted on the role of *ET*, which is an especial important omission as climate change will affect these northern latitude ecosystems more strongly than temperate and tropical biomes. Additionally, more references about the study location will be provided, which will help guide the reader to find additional studies that have previously used this well-established study site.

Reviewer: it is a stylistic choice to consider ET as a "loss", I might vary the term more and use flux ? Who loses ? Couldn't it also be seen as a positive flux for the ecosystem and the atmosphere? If indeed you want to maintain this language so strongly, you need to define your balance equation early on in terms of what is "positive" and what is "negative".

*Authors*: We agree that *ET* is a water flux and that it may be misleading, and potentially confusing, to consider *ET* as a "loss". We will therefore carefully go through the manuscript and replaced "loss" with *ET* and its component fluxes. We will also rephrased the first sentence in the introduction to now describe the movement of water in terrestrial ecosystems as inputs and outputs.

Reviewer: you mention groundwater recharge, carbon cycle, stream flow, but you never come back to answer any applied question regarding water use

*Authors*: We mentioned streamflow, groundwater recharge and ecosystem carbon cycle at the end of the first paragraph in the introduction to highlight the important role that ET has on other water balance components as well as the ecosystem carbon cycle. In doing so, we are stressing the importance of understanding the magnitude and drivers of ET, which is the focus of the manuscript. In the discussion, we compare the magnitude of ET to stream flow as well as discuss how a future climate could affect the overall water budget and carbon balance in boreal forests.

Reviewer: "quantifying the magnitude and drivers of transpiration and evaporation are crucial to better understanding the spatiotemporal variation of water fluxes in terrestrial ecosystems." => do you do this in your discussion?

Authors: We acknowledge that this study does not directly investigate the abiotic drivers (i.e., environmental factors) influencing ET and its different flux components (i.e., transpiration and evaporation). However, this was beyond the scope of the current study and is something we will address in a separate manuscript. The focus of this study was to evaluate the closure of the water budget, which is rarely done in a single study, and compare the magnitude of different water balance components during the growing season. We will rephrase this sentence to make this clear. In the discussion, we highlight the relative importance of transpiration and evaporation as well as compare these vertical water fluxes to measured horizontal water fluxes (i.e., stream runoff). In doing so, we believe that this study provide a better understanding of how water moves in the boreal forested landscape during the growing season.

Reviewer: the review of ET partitioning research and ET in general seems quite limited. Stable isotopes are mentioned as the only tool that is not used in this paper, while there are others. I believe there are quite a few reviews that discuss the history of evaporation and evapotranspiration research out there, for example Gabriel Katul et al. 2012: EVAPOTRANSPIRATION: A PROCESS DRIVING MASS TRANSPORT AND ENERGY EXCHANGE IN THE SOIL-PLANT-ATMOSPHERE-CLIMATE SYSTEM in the Reviews of Geophysics. This gives a history of evaporation and transpiration research which starts much earlier than when you claim the interest started.

Authors: We agree that there is a rich history of evaporation and evapotranspiration research and will add a few sentences in the introduction to highlight this history. We will also delete the sentence that mentions how stable isotopes can be used to partition ET as this approach is not used in our study and is just one example of the many different approaches that can be used to partition ET.

Reviewer: Your entire approach neglects ground water recharge and deep soil leakage. Perhaps this is negligible at your site, but you need to address this. I believe that presenting your water balance equation in the introduction would help you articulate the assumptions. Additionally are bare ground and open water 100% non existent in your site? You need to address that there might be spatial variations in evaporation and transpiration and that your measurements are still at the point scale. You mention that the vegetation cover is homogeneous, but is it that homogeneous that you can ignore spatial variation?

Authors: We will add a paragraph in the methods section that describes how we partitioned the water balance based on empirical measurements. In doing so, we now include information on how we calculated "ground water recharge" and/or "deep soil leakage", which is what we call "ds/dt" in Figure 5.

We will provide a high resolution (17 cm2 pixels) aerial photo of the C2 subcatchment and surrounding area to more clearly show the homogenous vegetation cover at our site. Additionally, when describing the study site we will now mention that that C2 subcatchment

is (i) completely covered (99.9 % canopy cover; Laudon et al. 2013) with a mixed forest stand (ii) there is no bare ground, and (iii) aside from the small (< 0.5 m wide) headwater stream, there is no open water.

Reviewer: Why only 1 growing season ?  And why only growing season?  I understand there are limitations based on available data  / instruments etc.  But I still think you should argue why this time frame is relevant for your questions and tell us what happens the rest of the year.  I suppose you have some measurements the rest of the year.

*Authors*: We acknowledge that it is unfortunate that our study period is only one growing season, but this this is the time period in which all measurements were available, namely empirical measurements of canopy transpiration and evaporation of precipitation from canopy trees. However, previous work from a nearby site has shown that there is little vegetation activity during the winter months (Tor-Ngern et al., 2017) and we have previously shown that evaporation of intercepted snow in the tree canopy represents 30% of winter precipitation at our site (Kozii et al., 2017). Partitioning the water balance during the growing season is more complicated, as trees are actively transpiring water during this time period resulting in a major water flux pathway that needs to be understood in more detail. Consequently, we know less about the movement of water in boreal forest during the active growing season, which is the focus of this study.

We will include daily hydro-meteorological variables for the entire 2016 year in Figure 2, to more clearly show the strong seasonality in stream runoff and environmental conditions affecting transpiration (*i.e.*, freezing temperatures). We will also include a sentence in the discussion section that highlights our previous work on evaporation of intercepted snow in canopy trees and, in turn, shows that evaporation of intercepted precipitation in canopy trees is the largest ET flux component when expressed on an annual time scale.

Tor-Ngern, P., Oren, R., Oishi, A. C., Uebelherr, J. M., Palmroth, S., Tarvainen, L., Ottosson-Lofvenius, M., Linder, S., Domec, J. C., and Nasholm, T. (2017) Ecophysiological variation of transpiration of pine forests: synthesis of new and published results, Ecological Applications, 27, 118-133.

Kozii, N., Laudon, H., Ottosson-Lofvenius, M., and Hasselquist, N. J. (2017) Increasing water losses from snow captured in the canopy of boreal forests: A case study using a 30 year data set, Hydrological Processes, 31, 3558-3567.

Reviewer: Figure 1 - I think the map of Sweden would benefit from some latitude lines or the arctic circle. What is the shading of the C2 Map?  What is the elevation / variation?  How does land cover change? You say the altitude of the outlet but not of the highest point. Please provide some proof that the land cover is sufficiently homogeneous. Or discuss how it is not - for this is where your uncertainty come from. How many points are you measuring meteorological data at?  Put it on the map. Where was the EC station?  This should be on the map.

*Authors*: We will make changes to Figure 1 to better show: (*1*) where exactly this study was done (*i.e.*, showing the arctic circle), (*2*) the elevation and variation in elevation within the C2 subcatchment, (*3*) the homogenous forest cover within the C2 subcatchment, (*4*) the location of the ICOS tower, where eddy covariance and other meteorological measurements are being

made, (*5*) the location of the nodes where sap flow measurements are being made and (*6*) the location of where canopy throughfall was measured.

Reviewer: l 190:  write equations as their own line, it is hard to follow in text.

*Authors*: All equations will be written in their own line.

Reviewer:  if possible add TF to map

*Authors*: We will included the location of the throughfall measurements to the map in Figure 1.

Reviewer: there is some work on the statistics of measuring through fall with rain gauges, this might help you estimate the uncertainty.

*Authors*: We are a little confused with this comment. What we did was measure throughfall at 25 locations. We also characterized the canopy structure above each throughfall collector (*i.e.*, a two-meter horizontal distance for each collector) based on spatial canopy density data acquired from airborne laser scanning (ALS). We then looked at correlations between different canopy attributes and *IL*. We found that overall median height (ElevMADmedian) had the highest correlation with measured seasonal interception losses and could explain 77% of the variation in IL. To quantify the uncertainty of the event-based *IL* estimated from measurement, we grouped the 25 throughfall rain gauges into 5 groups based on the ElevMADmedian and calculate standard deviation for each group and event. The weighted average obtained as a result of the groupwise standard deviations (IL_STD) gave us an indication of the uncertainty of the *IL* for the entire C2 subcatchment. We will rewrite this section to make this clearer in the methods section.

Reviewer: write out how the weighting calculation was done, this isn't very clear.  Do you have a reference that TF is directly correlated with canopy density in this forest type?

*Authors*: Please see our response to the previous comment.

Reviewer: L205 put a table of those metrics and the correlations in the supplementary material.

*Authors*: The FUSION software provided us a total of 121 different canopy metrics. We assessed the correlation between *IL* and all 121 canopy metrics. We will include a new table in the supplementary material, but in this table we only present the 10 canopy metrics that had the highest correlation with seasonal *IL*.

Reviewer: through fall / interception is probably not as linear as you say.  Often it depends on rainfall intensity and wind etc.  I think showing this data would be a valuable contribution to the field and complement your paper.

*Authors*: We agree that throughfall/ interception for a single rain event cannot be predicted based solely on canopy metric, because as you mentioned throughfall/ interception also strongly depends on rainfall intensity, wind, etc. However, we want to stress that our estimates of *IL* are for the entire growing season and assessing how environmental factors (*i.e.*, rainfall intensity, wind speed, etc…) was beyond the scope of this study. However, in Figure 4b, c we present information on how rainfall intensity affect *IL*, and show a non-linear relationship between precipitation (*i.e.*, rainfall intensity) and *IL*. We now discuss how *IL* depends on rainfall intensity in the discussion section. Note that in the APES –model used in this work, rainfall interception depends on canopy storage capacity (linearly related to leaf area in the canopy layer), initial storage at the onset of rainfall event and rainfall intensity. The evaporation from wet leaves depends on microclimatic conditions (*i.e.*, wind, radiation, temperature etc.). Thus, in model simulations the rainfall frequency and intensity and temporal variations of weather conditions are accounted for.

Reviewer: Figure S2 could be on your primary map.

*Authors*: We will remove Figure S2 and now include the location of the nodes where sap flow was measured in the map of the C2 subcatchment in Figure 1

Reviewer: L 294 : solved ?

*Authors*: We have reworded this sentence. The sentence now reads as follows: "In our model, we used measured soil moisture and soil temperature at the depth of 0.05 m as lower boundary conditions."

Reviewer: L366 => T used a majority of available water.

*Authors*: We will rewrite this sentence to make it clear that transpiration was the largest water flux component during the study period.

Reviewer: L 374 => they include

*Authors*: We will replace "it includes" with "they include".

Reviewer: L 376 => observed data ?

*Authors*: We will rewrite this sentence to make it clear that we are comparing modeled estimates to measured data.

Reviewer: L 385 => area represents or areas represent

*Authors*: We will change "areas represents" to "areas represent".

Reviewer: figure 5 => incoming precipitation wasn't modeled, was it ? Understorey => Understory

*Authors*: Precipitation was not modelled. However, we understand how this could be misunderstood in Figure 5. We will place precipitation in the middle of the figure with arrows going to both the "measured" (left side) and "modelled" (right side) partitioning approaches. Additionally, we will use the abbreviation of *ET* flux components in the figure as suggested by reviewer #2.

Reviewer: L 438 - this is a long sentence

*Authors*: We have rewritten this sentence. The sentence now reads as follows: "Our findings clearly highlight the important role canopy trees play in the boreal hydrological cycle during the growing season, and stresses the need to better understand the effect of trees and their response to forest management practices and a changing climate."

Reviewer: comparing => compare

*Authors*: We have rewritten this sentence as suggested in the previous comment.

Reviewer: Maybe go one step further with a thought experiment. For example, based on our measurements, if all trees were removed from entire subcatchment, we would expect discharge over this period to go up by X%.

*Authors*: We agree that it is interesting to assess how changes in forest stand structure could influence the movement of water at our study site. Although we do not speculate how the removal of all trees would influence stream discharge, we do assess how changes in forest stand structure (i.e., leaf area index; LAI) could influence *ET* and its flux components. In the discussion section, we have a paragraph where we discuss the potential consequences of how changes in forest stand structure, as a result of forest management practices, could affect the way water moves through boreal forests.

Reviewer: this would have been interesting in introduction (with citations and further precision). In the introduction, could you have shown us an annual hydrograph and situation your study within that? (I know you showed a little bit of a hydrographic, but it wasn't the whole year).

*Authors*: We will change figure 2 to include environmental data and the hydrograph for the entire 2016 year. We will also include dotted vertical lines in the figure to clearly show the study period used in this study.

Reviewer: Figure 6 - It is hard to believe these are lines.

*Authors*: We are a little confused about this comment. In Figure 6, we present the results of model simulations in which we ran the APES model where LAI was changed from 1 to 7 m$^2$

m$^{-2}$ at 0.5 intervals. In doing so, we show how changes in LAI influences *ET* and it flux components. We explain this in the discussion section where we say: "*To assess how forest management practices may influence the overall magnitude of ET as well as the relative importance of the different ET flux components we ran the APES model with a range of canopy LAI values; from 1 to 7 m$^2$ m$^{-2}$.*" We will now place the two panels in Figure 6 next to each other to more clearly show how changes in LAI influences *ET* and its flux components.

---

## Author Comment (AC2) · 26 Feb 2020

Dear Miriam Coenders-Gerrits, thank you for your review and very constructive comments.

Main concerns:
Reviewer: L146: the author only consider the growing season. I think it's important to emphasis throughout the paper.

*Authors*: We agree that this is an important point to emphasis and will make sure to highlight this throughout the manuscript. We will also make a change to the title to emphasis that this study only considers the growing season. The new title will read as: "Partitioning growing season water balance within a forested boreal catchment using sapflux, eddy covariance and a process-based model"

Reviewer: L176: I am happy to see that below EC-system latent heat is considered. However, equation 1 is only valid once the forest is homogeneous since the footprint of the ECsystem and the below canopy latent heat are different. How 'homogeneous is your forest? Please elaborate. –

*Authors*: The C2 subcatchment is completely covered (99.9% forest cover) by an old growth (> 100 yr.) mixed forest stand. We will now include a high-resolution aerial photograph of the C2 subcatchment in Figure 1c that shows the homogenous forest cover within the subcatchment. We will also added some text in the methods section to better describe the homogenous nature of the forest stand within the C2 subcatchment.

Reviewer: L199: the TF-sampling was done on 'event-base'. Please elaborate on how this was done. Did you run into your forest after rain ceased? Or did you do daily observations? How did you defined 'an event'?

*Authors*: We collected water from individual rain gauges immediately after rain ceased and thus each rainstorm represents an 'event'. During the study period that corresponded to 26 rain events. We will add a sentence to the methods section that describes how water was collected from individual rain gauges immediately after each rain event and therefore estimates of *IL* were made on an event basis.

Reviewer: L207: How did you tested whether ALS had the highest correlation with seasonal interception loss?

*Authors*: We used the FUSION software to characterized the canopy structure above each throughfall collector (*i.e.*, a two-meter horizontal distance for each collector), which is based on spatial canopy density data acquired from airborne laser scanning (ALS). The FUSION software gave us a total of 121 different canopy metrics that describes the canopy structure. We then looked at the correlation coefficient between season IL for each of the 25 *TF* collectors and all 121 canopy metrics. We found that ElevMADmedian had the highest correlation with measured seasonal interception losses and could explain 77% of the variation in *IL*. We will add some text to the methods section to make this clear. We will also include a table in the supplementary material that show the 10 canopy metrics that had the highest correlation with seasonal *IL*, as suggested by reviewer #1.

Reviewer: L220-285: Please have a look at the recent technical note by Larsen et al 2019. Would it be necessary to compensate your sapflow measurements as well? Not doing this could mean an overestimation of your transpiration.

*Authors*: Thank you for bring this paper to our attention. The paper by Larsen et al. (2019) highlights the concerns of probe misalignment when using heat pulse sensors for sap flow measurements. In our study, we used the heat dissipation approach and it is unclear if probe misalignment has the same effect, or has any effect, and if it has an effect whether the proposed correction based on heat pulse sensors would work for heat dissipation sensors. Employing the correction therefore may increase the error.

In our study we accounted for known sources of variation associated with radial, azimuthal and trees size in an attempt to minimize errors association with our calculations of transpiration. Although we employed the same coefficients when calculating transpiration we believe this has a minimal effect because the approach we used has previously been shown to produce reasonable results, especially in conifers, based on comparisons with eddy covariance and mass balance approaches (Oren et al. 1998; Schäfer et al. 2002; Ward et al. 2008; Oishi et al 2008; Tor-ngern et al. 2018; Ward et al. 2018)).

Oren R, Phillips N, Katul G, Ewers BE, Pataki DE (1998) Scaling xylem sap flux and soil water balance and calculating variance: a method for partitioning water flux in forests. *Annales des Sciences Forestieres* 55:191-216

Schäfer KVR, Oren R, Lai CT, Katul GG (2002) Hydrologic balance in an intact temperate forest ecosystem under ambient and elevated atmospheric $CO_2$ concentration. *Global Change biology* 8: 895-911

Ward EJ, Oren R, Sigurdsson BD, Jarvis PG, Linder S (2008) Fertilization effects on mean stomatal conductance are mediated through changes in the hydraulic attributes of mature Norway spruce trees. *Tree Physiology* 28: 579-596.

Oishi AC, Oren R, Stoy PC (2008) Estimating components of forest evapotranspiration: A footprint approach for scaling sap flux measurements. *Agricultural and Forest Meteorology* 148: 1719-1732

Tor-ngern P, Oren R, Palmroth S, Novick K, Oishi A, Linder S, Ottosson-Löfvenius M, Näsholm T (2018) Water balance of pine forests: synthesis of new and published results. *Agriculture and Forest Meteorology* 259:107-117

Ward EJ, Oren R, Kim HS, Kim D, Tor-ngern P, Ewers BE, McCarthy HR, Oishi AC, Pataki DE, Palmroth P, Phillips NG, Schäfer KVR (2018) Evapotranspiration and water yield of a pine-broadleaf forest are not altered by long-term atmospheric [$CO_2$] enrichment under native or enhanced soil fertility. *Global Change Biology* 24: 4841-4856. DOI: 10.1111/gcb.14363

Reviewer: section 2.3: a better explanation of the modelling principles of APES, would help the reader. For example showing model-scheme.

*Authors*: We will reorganize and streamline Section 2.3 to provide a better overview of the modeling principles of APES. The reader can find a Figure of the model scheme in Launiainen et al. (2015), which we cite when describing the model.

Launiainen, S., Katul, G. G., Lauren, A., and Kolari, P. (2015) Coupling boreal forest $CO_2$, $H_2O$ and energy flows by a vertically structured forest canopy – Soil model with separate

bryophyte layer, Ecological Modelling, 312, 385-405.

Reviewer: section 2: I think it would help to make a schematic picture (a bit like figure 5) of how you define ET and its subcomponents.

*Authors*: We acknowledge that it is a little unclear on how exactly we define and quantify *ET* and its flux components. We will therefore add a paragraph to the beginning of section 2 that clearly explains how we calculated *ET* and it individual flux components. We could also include a schematic picture if it is deemed necessary.

Reviewer: L376-380: be careful with your definitions of transpiration, evaporation and evapotranspiration. ETu is a combination of forest floor interception, understory transpiration (mosses) and soil evaporation and is thus not only 'evaporation' as said in L378. Also the role of soil evaporation is not explained. Is soil evaporation relevant in your study site? Why/why not.

*Authors*: We have carefully gone through the manuscript to make sure we are consistent with our definitions of transpiration, evaporation, and evapotranspiration. Additionally, we will change *IL* to *Ic* throughout the manuscript to make it clear that we are talking about evaporation of intercepted precipitation in the tree canopy. In this specific case (L376-380), we will rephrase this sentence to be clear that we are talking about *Ic* and understory evapotranspiration (*ETu*).

At our site, soil evaporation is negligible as there is no bare ground within the C2 subcatchment. We will make this clear in the method section when describing the study site by stating that the understory consists of a continuous layer of bilberry (Vaccinium myrtillus), lingonberry (Vaccinium vitis-idaea), and mosses (Pleurozium schreberi and Hylocomium splendens) with no bare ground."

Reviewer: Section 4: the discussion and conclusions are merged into one section. I think it would be better to split this. And/or merge the discussion with the results section. But for sure make a separate section for the conclusions only where you are only answering to the research objective.

Authors: We will make a separate section for the conclusions.

Specific (minor) comments:
Reviewer: L31: redundant to mention "and being roughly 7 times greater than stream runoff". This is the same info as saying ET is 85

*Authors*: We will remove "and being roughly 7 times greater than stream runoff" from the sentences.

Reviewer:  L44: Maybe better to mention the spread in global ET. This is ca 55-80

*Authors*: We will now include the spread in global *ET*.

Reviewer: L71: after e.g. a comma.

*Authors*: We will add a comma after e.g.

Reviewer: L128: unit of annual rainfall is mm/year.

*Authors*: We will include yr$^{-1}$ in our units of annual rainfall.

Reviewer: L157-165: variables like P, Q, dS, etc should be in italic.

*Authors*: We will italicized all water balance components (i.e., *P*, *Q*, *ET*, *T*, *IL*, *ETu*, and *ds/dt*) in this section and throughout the manuscript.

Reviewer: L165: I prefer to rename dS into dS/dt, since dS is the storage change per time.

*Authors*: We will change ΔS to ds/dt in this section as well as throughout the manuscript.

Reviewer: L172: details => detail.

*Authors*: We have rephrased this sentence as suggested by reviewer #3. The sentence now reads as follows: "A detailed description of the EC data processing and quality control can be found in Chi et al. (2019)"

Reviewer: Fig S1: the unit of P is mm/y. Furthermore, I would change instead of showing Q/P, showing ET/P. Since this the focus of the paper.

*Authors*: We will change the units of P to mm yr$^{-1}$ as well as now show ET/P in Figure S1.

Reviewer: L337-342: This is a result.

*Authors*: We agree that L337-342 can be interpreted as a result, but we consider this finding as a test of the validity of the model at our study site. As the APES model was able to represent individual components of the surface energy balance reasonably well, it gives us confidence on the model's predictions of *ET* and its flux components. This information is only used as a model check and thus we choose to present it in this section and as a supplementary figure.

Reviewer: Fig3c: why showing IL+ETu? Why not only ETu? This would more sense in my view.

*Authors*: We agree that it would be nice to directly compared daily values of "measured" and "modeled" *ETu* during the study. However, this was not possible because canopy interception loss ($I_C$) were determined on an event-basis, and not on a daily basis. The "measured" data presented in Figure 3c is the difference between *ET* and canopy transpiration, which is $I_C$ + *ETu*. We will rewrite the figure caption to make this clearer.

Reviewer: Section 2/fig 3: explain how ETu is 'measured'. It's calculated as ETu=ET-IL-T, right? Please add this equation and elaborate on the fact that ETu is thus not independent of the other measured components.

*Authors*: Understory evapotranspiration (*ETu*) was not directly measured in this study, but instead was calculated as: $ETu = ET – I_C – T$. Moreover, because $I_C$ was estimated on an event basis, our estimate of *ETu* was for the entire growing season. We will add text in the method section that better describes how *ETu* was calculated.

Reviewer: Figure 5: I would add the percentages as well. Furthermore, be consistent in the naming of ET and its subcomponents. Would it not be better to use here the abbreviations?

*Authors*: In Figure 5, we now include the percentage of individual flux in relation to total *ET*. We did not include the percentage of individual flux components in relation to incoming *P*, as we believe this may cause confusion and would make the figure more difficult to understand. However, the values of total *P* and individual water pathways are presented in this figure, which makes it possible to also determine the percentage of different water pathways in relation to total *P*. Additionally, we now use the abbreviation for the different *ET* flux components in Figure 5.

---

## Author Comment (AC3) · 26 Feb 2020

Dear Anonymous Reviewer, thank you for your review and very constructive comments.

Reviewer: The word composition of the title is not clear " ´ . . .forest water balance. . ." is it partitioning of water balance in boreal forest, or partitioning forest-water balance?

*Authors*: We have changed the title to: "Partitioning growing season water balance within a forested boreal catchment using sapflux, eddy covariance and a process-based model".

Abstract:
Reviewer: It would be nice to see water balance ways more specific to boreal ´ forests to get a clearer picture how this work is worthy for readers

*Authors*: In the abstract, we will make it clear that few studies have partitioned ET into it individual flux components in boreal forests. Also, in the introduction we will highlight the considerable variation in the relative importance of ET in boreal forests, ranging between 45-85 % of incoming P. Thus, quantifying the magnitude and spatiotemporal variation in transpiration and evaporation is crucial to better understand ET and its importance in boreal forests.

Reviewer:  In line 20, it ´ reads "water is lost"; this is very confusing wording all over the paper. 1) water cannot be lost from a system, 2) I assume this paper deals with water balance, so water "flows" from one state/regime to next, and that is not lost, 3) there could be some cases where ET can be referred as lost; that is when rainfall is dealt as "gain"

*Authors*: We agree that ET is a water flux and that it may be misleading, and potentially confusing, to consider ET as a "loss".  We will therefore carefully go through the manuscript and replaced "loss" with ET and its component fluxes and no longer refer to ET as a "water loss"

Reviewer: Line ´ 30 change "water loss pathway" to "water balance component"

*Authors*: We will change "water loss pathway" to "water balance components".

Reviewer: Line 32 Canopy ´ interception is not part of ET, it should be rather evaporation from canopy

*Authors*: We agree that interception in not part of ET, but rather evaporation of intercepted water in canopy trees. We will rewrite this sentence to make this clear.

Reviewer: Line 33- ´ 34, the numbers do not add up 70, check

*Authors*: We agree that the numbers in line 33-34 do not add up to 70. However, the number presented in lines 33-34 represented the percentage of T and IL to total ET, whereas the 70 % is in reference to T and IL being equal to ca. 70 % on the incoming precipitation during the growing season. We will rewrite this sentence to make this clear.

Introduction:
Reviewer:  The study has got no clear ´ definition of hypothesis or purpose of the study

*Authors*: The objectives of the study are stated in the final paragraph of the introduction: The main objective of this study was to *i*) constrain the absolute and relative magnitudes of *ET* flux components by using both empirical data and model simulations, *ii*) to explore how they vary during the course of the growing season, *iii*) to compare different ET flux components to other water balance components (*i.e.*, stream runoff) and *iv*) directly assess the important role trees play in the boreal hydrological cycle during the growing season.

Reviewer: Line 51-52, I don't agree that most ´ studies treat ET as a single water flux pathway

*Authors*: We will remove this sentence from the manuscript.

Reviewer: Line 62-63, I think, rather there ´ are dozens of experimental studies for decades

*Authors*: We will rewrite this sentence to acknowledge the long history of research on ET as suggested by reviewer #1 as well as highlight the number of different approaches and methodology to partition ET into its individual flux components, which includes numerous empirical measurements as well as modeling approaches.

Reviewer: Line 73, what does it mean by ´ "few investigation on water balance at catchment scale"?

*Authors*: We are trying to highlight that the majority of ET partitioning studies have been done at the stand and/or plot scale and thus are not able to directly compare the magnitude of ET and its flux components to other water pathways (*i.e.*, steam runoff). We will rewrite this sentence to make this clearer.

Reviewer: The paragraph after line ´ 90 better fits above the previous paragraph

Authors: We agree and will move this section to the previous paragraph.

Reviewer: Line 114, what is the state-of-the-art ´ of hydrological measurements at the study site? Give some details of measurements done which of course respective to this study

*Authors*: We are trying to highlight that this study builds upon the rich history of long-term hydrological measurements within the Krycklan catchment. We will remove "state-of-the-art" from this sentence and make this point clearer.

Methods:
Reviewer: Line 147-148, not clear ´

*Authors*: We will remove "spanning from after the spring flood until leaf senescence for deciduous species" from the sentence.

Reviewer: Line 153-155, not clear

*Authors*: We will remove this sentence from the manuscript.

Reviewer: Line 157, what are the environmental data, give the ´ details or examples

*Authors*: We will provide details about the instruments used to measure environmental data.

Reviewer: Paragraph line 165-175, Too much information. Please classify ´ with instruments, data, how processed, calibrated, purpose – this might help readers to understand

*Authors*: We will reorganize and streamline the description of the eddy covariance measurements as suggested.

Reviewer: Line 179, what does it mean by "non-stationarity" this word commonly ´ used in statistical description not in instrumentation

*Authors*: We will rewrite this sentence to more clearly describe how the ET data was filtered using the EddyPro quality check and flagging policy. More specifically, we will replace "non-stationarity" with "tests on steady state".

Reviewer:  Assumptions described in line ´ 188-190 are wrong, re-write (it should be IL = GP-TF-SF)

*Authors*: We are aware that stemflow (SF) is often included when calculating canopy interception losses (i.e., IL = GP – TF – SF). However, previous work within the Krycklan catchment has shown no SF in forest stands dominated by spruce and pine trees during the summer months (Venzke, 1990). Thus, we have omitted SF when calculating IL in our study. We will add a sentence in the methods sections that highlights this previous observation which in turn provides justification for our calculation of IL as the difference between GP and IL.

Venzke, J. F. (1990) Beiträge zur Geoökologie der borealen Landschaftszone. Geländeklimatologische und pedologische Studien in Nord-Schweden, Verlag Ferdinand

Schöningh, Paderborn, Germany.

Results and discussion

Reviewer: Are mixed up and not well structured: please take rendering sentences from results to discussion

*Authors*: We will carefully go through the results and discussion section to better improve its structure as well as make sure that all interpretation of the data is moved to the discussion section.

---

## Author Response (AR1)

Dr. Ryan Teuling
Editor
Hydrology and Earth System Sciences

RE: hess-2019-541

Dear Dr. Ryan Teuling

Thank you for giving us the opportunity to address the reviewers' comments. We believe we have been able to satisfactorily address all substantive comments which have further improved the manuscript. Detailed responses to reviewer's comments are described below. We reference the manuscript with track changes when describing specific lines where changes were made.

**Reviewer #1: Dr. Natalie Ceperley**

- Although the word boreal is mentioned, we aren't introduced to the specifics of why it matters or a clear definition, situation on the globe.
We have added a few sentences to the introduction that highlights: (i) the extent of the boreal forest (*i.e.*, one of the largest biomes in the world), (ii) the important role boreal forests play in the global water and carbon cycles as well as global climatology and (iii) the few studies that have shown large variation in the partitioning of the water balance in these ecosystems (i.e., *ET* represent 45–85 % of incoming *P* in boreal forests). The added text reads as follows: "*Boreal forests cover ca. 12 million km² of land area and represents the second largest biome behind tropical forests (Bonan, 2008). Given their large size, boreal forests regulate water and energy fluxes over a vast area and thus play an important role in global hydrology and climatology (Bonan, 2008;Baldocchi et al., 2000;Chen et al., 2018). Boreal forests also play an important role in the global carbon cycle (Goodale et al., 2002); sequestering ca. 0.5 petagrams of carbon annually and storing approximately one third of the global terrestrial carbon (Bradshaw and Warkentin, 2015;Pan et al., 2011). However, few studies have partitioned the water balance in boreal forests (Talsma et al., 2018;Peel et al., 2010;Tor-ngern et al., 2018). In the ones that have, ET has been shown to represent 45-85% of incoming P (Peel et al., 2010).*" (L 51-60)

- As someone who has never worked in this part of the world, I have never heard of Krycklan and the term "boreal" does not automatically invoke an idea of what the issues / steaks / interest is regarding the water balance and evapotranspiration. The location is an extremely valuable part of this research and story, please introduce it well.
Similar to the answer above we have now added text about the boreal biome, but also better highlight the high latitude location of the study site. This is a geographic area where relatively much less research has been conducted on the role of ET, which is an especial important omission as climate change will affect these northern ecosystems more strongly than the temperate and tropical biomes. We have also added additionally information about the study location to now guide the reader to find more literature that has previously used this well studied catchment.

- it is a stylistic choice to consider ET as a "loss", I might vary the term more and use flux ? Who loses ? Couldn't it also be seen as a positive flux for the ecosystem and the atmosphere?

If indeed you want to maintain this language so strongly, you need to define your balance equation early on in terms of what is "positive" and what is "negative".

We agree that ET is a water flux and that it may be misleading, and potentially confusing, to consider ET as a "loss". We have therefore carefully gone through the manuscript and replaced "loss" with ET and its component fluxes. We have also rephrased the first sentence in the introduction to now describe the movement of water in terrestrial ecosystems as inputs and outputs. The sentence now reads as follows: "*In the hydrological cycle, water enters terrestrial ecosystems mainly through precipitation (P). This water leaves terrestrial ecosystems either through evapotranspiration (ET) back to the atmosphere or as stream runoff (Q).*" (L41-43)

- you mention groundwater recharge, carbon cycle, stream flow, but you never come back to answer any applied question regarding water use

We mentioned streamflow, groundwater recharge and ecosystem carbon cycle at the end of the first paragraph in the introduction to highlight the important role that ET has on other water balance components as well as the ecosystem carbon cycle. In doing so, we are stressing the importance of understanding the magnitude and drivers of ET, which is the focus of the manuscript.

In the discussion, we compare the magnitude of ET to stream flow as well as discuss how a future climate could affect the overall water budget as well as the carbon balance in boreal forests. This section in the discussion reads as follows: "*Within the Krycklan Catchment, roughly 40 % of annual stream runoff occurs as a response to snowmelt (Ågren et al., 2012), when trees are relatively inactive (Tor-Ngern et al., 2017). In this study, we found that ET becomes the dominant water flux after spring flood has ceased, and during the growing season it was seven times greater than stream runoff (Fig. 2c, 3a). In our study, combining P with modelled estimates of ET and measured stream runoff results in a negative water balance (P < ET + Q) during the growing season. This is in agreement with other studies in boreal forests, which have found a negative water balance during the growing season (Wang et al., 2017;Tor-ngern et al., 2018;Sarkkola et al., 2013). Such asynchrony in the relative importance of different water balance components might be even more pronounced in a future climate when higher air temperatures and less frequent, albeit more intense, precipitation events can be expected (IPCC, 2018). One future scenario is earlier snow melt and less snow accumulation during winter as a result of higher air temperatures (Byun et al., 2019), which would result in earlier peak stream runoff thereby reducing the annual amount of water available for tree growth during the growing season (Barnett et al., 2005). This, in turn, could have cascading effects on forest productivity (Barber et al., 2000;Silva et al., 2010), tree mortality (Peng et al., 2011) and the overall carbon balance in boreal forests (Ma et al., 2012).*" (L492-516)

- "quantifying the magnitude and drivers of transpiration and evaporation are crucial to better understanding the spatiotemporal variation of water fluxes in terrestrial ecosystems." => do you do this in your discussion?

We acknowledge that this study does not directly investigate how abiotic drivers (*i.e.*, environmental factors) influence ET and its different flux components (i.e., transpiration and evaporation). However, this was beyond the scope of the current study and something we will address in a separate manuscript. The focus of this study was to evaluate the closure of the water budget, which is rarely done in a single study, and compare the magnitude of different water balance components during the growing season. We have rephrased this sentence to make this clear. The sentence now reads as follows: "*Thus, quantifying the magnitude and*

*spatiotemporal variation of T and evaporation separately is crucial to better understanding how water moves through boreal forest landscapes.*" (L69-71)

In the discussion, we highlight the relative importance of transpiration and evaporation as well as compare these vertical water fluxes to measured horizontal water fluxes (*i.e.*, stream runoff). If doing so, we believe that this study provides a better understanding of how water moves in the boreal forested landscape during the growing season.

- the review of ET partitioning research and ET in general seems quite limited.  Stable isotopes are mentioned as the only tool that is not used in this paper, while there are others.  I believe there are quite a few reviews that discuss the history of evaporation and evapotranspiration research out there, for example Gabriel Katul et al. 2012: EVAPOTRANSPIRATION: A PROCESS DRIVING MASS TRANSPORT AND ENERGY EXCHANGE IN THE SOIL-PLANT-ATMOSPHERE-CLIMATE SYSTEM in the Reviews of Geophysics.  This gives a history of evaporation and transpiration research which starts much earlier than when you claim the interest started.

We agree that there is a rich history of evaporation and evapotranspiration research and have added a few sentences in the introduction to highlight this history. We also deleted the sentence that mentions how stable isotopes can be used to partition ET as this approach is not used in our study and is just one example of the many different approaches that can be used to partition ET. The section now reads as follows: "*Research investigating the biotic and abiotic controls on ET has a long history, dating back centuries (Katul et al., 2012;Brutsaert, 1982). However, efforts to separately estimate T and evaporation began in the 1970s (see Kool et al., 2014) and ever since there has been an increasing number of studies partitioning ET (Stoy et al., 2019;Schlesinger and Jasechko, 2014). There are a number of different approaches and methodology to partition ET into its individual flux components (Kool et al., 2014), including empirical measurements (Mitchell et al., 2009;Cavanaugh et al., 2011;Good et al., 2014;Sutanto et al., 2014) as well as a number of different process based models (Sutanto et al., 2012;Stoy et al., 2019;Launiainen et al., 2015). Each of these different approaches have their advantages and disadvantages and it has been shown that the relative contribution of different ET flux components differs depending on the approach used (Schlesinger and Jasechko, 2014). It has therefore been highlighted that the use of multiple methods is desirable to more accurately partition ET into it individual flux components (Stoy et al., 2019).*" (L72-84)

- Your entire approach neglects ground water recharge and deep soil leakage.  Perhaps this is negligible at your site, but you need to address this.  I believe that presenting your water balance equation in the introduction would help you articulate the assumptions.  Additionally are bare ground and open water 100% non existent in your site?  You need to address that there might be spatial variations in evaporation and transpiration and that your measurements are still at the point scale.  You mention that the vegetation cover is homogeneous, but is it that homogeneous that you can ignore spatial variation?

We have now added a paragraph in the methods section that describes how we partitioned the water balance based on empirical measurements. In doing so, we now include information on how we calculated "ground water recharge" and/or "deep soil leakage", which is what we call "*ds/dt*" in Figure 5. The added paragraph reads as follows: "*We used the hydrological mass balance approach in combination with empirical measurements of vertical and horizontal water fluxes to quantify the water balance components within the C2 subcatchment. The mass balance equation is*

$$ds/dt = P - ET - Q \qquad (1)$$

*where ds/dt is change in soil water storage per unit area and Q is stream runoff. ET was measured using the eddy covariance technique, and partitioned into components as*

$$ET = T + I_C + ET_U \qquad (2)$$

*where canopy tree T was determined using sap flow sensors and evaporation of intercepted P from the tree canopy ($I_C$) was determined as the difference between open sky precipitation and water collected on event basis in rain gauges placed below the canopy (see below). Understory evapotranspiration (ETu) was not directly measured in this study, but was instead calculated as*

$$ET_U = ET - I_C - T \qquad (3)$$

*Because $I_C$ was estimated on an event basis, our estimate of ETu was for the entire growing season. Daily stream runoff (Q) was calculated as daily discharge, obtained from the Svartberget data portal (https://franklin.vfp.slu.se/), per catchment area. Change in soil water storage (ds/dt), which includes ground water recharge, was calculated as the residual of the hydrological mass balance (eq. 1).*" (L167-184)

We also now provide a high resolution aerial photo of the C2 subcatchment and surrounding area to more clearly show the homogenous vegetation cover at our site. Additionally, when describing the study site we now mention that the C2 subcatchment is (i) completely covered (99.9% canopy cover; Laudon et al. 2013) with a mixed forest stand (ii) there is no bare ground and (iii) aside from the small (< 0.5 m wide) headwater stream, there is no open water. The description of our study site now reads as follows: "*The C2 subcatchment is completely covered by an old growth (>100 yr.) mixed forest stand of Picea abies (61 %), Pinus sylvestris (34 %), and Betula (5 %) (Laudon et al. 2013). The understory consists of a continuous layer of bilberry (Vaccinium myrtillus), lingonberry (Vaccinium vitis idaea), and mosses (Pleurozium schreberi and Hylocomium splendens) with no bare ground. Aside from the small (< 0.5 m wide) headwater stream, there is no open water within the C2 subcatchment. Similar forest stands extend to the east and west of the C2 subcatchment boundaries by several hundred meters (Fig. 1c).*" (L149-155)

Why only 1 growing season ?  And why only growing season?  I understand there are limitations based on available data  / instruments etc.  But I still think you should argue why this time frame is relevant for your questions and tell us what happens the rest of the year.  I suppose you have some measurements the rest of the year.
We acknowledge that it is unfortunate that our study period is only one growing season, but this this is the time period in which all measurements were available, namely empirical measurements of canopy transpiration and evaporation of precipitation from canopy trees. However, previous work from a nearby site has shown that there is little vegetation activity during the winter months (Tor-Ngern et al., 2017) and we have previously shown that evaporation of intercepted snow in the tree canopy represents 30% of winter precipitation at our site (Kozii et al., 2017). Partitioning the water balance during the growing season is more complicated, as trees are actively transpiring water during this time period resulting in a major water flux pathway that needs to be understood in more detail. Consequently, we know less about the movement of water in boreal forest during the active growing season, which is the focus of this study.

We have now included daily hydro-meteorological variables for the entire 2016 year in Figure 2, to more clearly show the strong seasonality in stream runoff and environmental conditions affecting transpiration (*i.e.*, freezing temperatures). We have also added a few sentences in the discussion section that highlights our previous work on evaporation of intercepted snow in canopy trees and, in turn, shows that evaporation of intercepted precipitation in canopy trees is the largest ET flux component when expressed on an annual time scale. The added section reads as follows: "*In a previous study at the Krycklan catchment, we found that evaporation of intercepted snow in the tree canopy represents ca. 30 % of winter (November – March) precipitation (Kozii et al., 2017). Thus, $I_C$ represents the largest ET component when expressed on an annual time scale as there is negligible T during the winter months (Tor-Ngern et al., 2017).*" (L542-546)

Tor-Ngern, P., Oren, R., Oishi, A. C., Uebelherr, J. M., Palmroth, S., Tarvainen, L., Ottosson-Lofvenius, M., Linder, S., Domec, J. C., and Nasholm, T. (2017) Ecophysiological variation of transpiration of pine forests: synthesis of new and published results, Ecological Applications, 27, 118-133.

Kozii, N., Laudon, H., Ottosson-Lofvenius, M., and Hasselquist, N. J. (2017) Increasing water losses from snow captured in the canopy of boreal forests: A case study using a 30 year data set, Hydrological Processes, 31, 3558-3567.

Figure 1 - I think the map of Sweden would benefit from some latitude lines or the arctic circle. What is the shading of the C2 Map? What is the elevation / variation? How does land cover change? You say the altitude of the outlet but not of the highest point. Please provide some proof that the land cover is sufficiently homogeneous. Or discuss how it is not - for this is where your uncertainty come from. How many points are you measuring meteorological data at? Put it on the map. Where was the EC station? This should be on the map.
We made changes to Figure 1 to better show: (*1*) where exactly this study was done (i.e., showing the arctic circle), (*2*) the elevation and variation in elevation within the C2 subcatchment, (*3*) the homogenous forest cover within the C2 subcatchment, (*4*) the location of the ICOS tower, where eddy covariance and other meteorological measurements are being made, (*5*) the location of the nodes where sap flow measurements are being made and (*6*) the location of where canopy throughfall was measured. The caption of Figure 1 now reads as follows: "***Figure 1.*** *Location of the study area in northern Sweden. (a) The outline of Sweden with the location of the Arctic Circle for reference. (b) The boundary of the 68 km² Krycklan Catchment with various subcatchment in different color; C2 subcatchment in yellow. Throughfall (TF) measurements were made ca. 1 km from the C2 subcatchment and are shown on this map (blue circle). (c) High resolution aerial photograph with five-meter contour intervals (white line) and the C2 subcatchment boundary (yellow line). Sap flow measurements were made at three nodes (green circles) and all environmental and eddy-covariance data were taken from the ICOS tower (yellow circle). (d) Picture of the forest stand with understory vegetation that is characteristic of the C2 subcatchment.*" (L139-147)

l 190: write equations as their own line, it is hard to follow in text.
We have now written all equations in their own separate line.

- if possible add TF to map
We now included the location of the throughfall measurements to the map in Figure 1.

- there is some work on the statistics of measuring through fall with rain gauges, this might help you estimate the uncertainty.
We are a little confused with this comment. What we did was measure throughfall at 25 locations. We also characterized the canopy structure above each throughfall collector (*i.e.*, a two-meter horizontal distance for each collector) based on spatial canopy density data acquired from airborne laser scanning (ALS). We then looked at correlations between different canopy attributes and IL. We found that overall median height (ElevMADmedian)

had the highest correlation with measured seasonal interception losses and could explain 77% of the variation in $I_C$. To quantify the uncertainty of the event-based $I_C$ estimated from measurement, we grouped the 25 throughfall rain gauges into 5 groups based on the ElevMADmedian and calculate standard deviation for each group and event. We have rewritten this section in the methods to make this clear. The section now reads as follows: "*Measurements of TF were made between the beginning of July and the end of October 2016. Water was collected from individual rain gauges immediately after each rain event resulting in event-based $I_C$ estimates (Gash, 1979). Spatial canopy density data acquired from airborne laser scanning (ALS) was used in the FUSION software (McGaughey, 2012) to characterized the canopy structure above each throughfall collector (2 m radius around each collector). We found that the absolute deviation of ALS height measurements from overall median height (ElevMADmedium) showed the highest correlations to $I_C$ and could explain 77% of variation in seasonal $I_C$ (Table S1). $I_C$ within the C2 subcatchment was estimated as a weighted averages of the 25 throughfall collector. The weighting was based on the ElevMADmedium around each throughfall collector and the frequency distribution of this metric within the entire C2 subcatchment. To quantify the uncertainty of event-based $I_C$, we grouped throughfall collectors into five groups based on ElevMADmedium and calculated the standard deviation for each group and event.*" (L226-238)

- write out how the weighting calculation was done, this isn't very clear. Do you have a reference that TF is directly correlated with canopy density in this forest type?
Please see our response to the previous comment.

- L205 put a table of those metrics and the correlations in the supplementary material.
The FUSION software provided us a total of 121 canopy matrices. We assessed the correlation between $I_C$ and all 121 canopy matrices. We now include a table in the supplementary material (Table S1) that shows the 10 canopy matrices that had the highest correlation with seasonal $I_C$.

- through fall / interception is probably not as linear as you say. Often it depends on rainfall intensity and wind etc. I think showing this data would be a valuable contribution to the field and complement your paper.
We agree that throughfall/ interception for a single rain event cannot be predicted based solely on canopy metric, because as you mentioned throughfall/ interception also strongly depends on rainfall intensity, wind, etc. However, we want to stress that our estimates of $I_C$ are for the entire growing season and assessing how environmental factors (*i.e.*, rainfall intensity, wind speed, etc…) was beyond the scope of this study. However, in Figure 4b, c we present information on how rainfall intensity affect $I_C$, and show a non-linear relationship between precipitation (*i.e.*, rainfall intensity) and $I_C$. Note that in the APES –model used in this work, rainfall interception depends on canopy storage capacity (linearly related to leaf area in the canopy layer), initial storage at the onset of rainfall event and rainfall intensity. The evaporation from wet leaves depends on microclimatic conditions (*i.e.*, wind, radiation, temperature etc.). Thus, in model simulations the rainfall frequency and intensity and temporal variations of weather conditions are accounted for.

We now discuss how $I_C$ depends on rainfall intensity in the discussion section. The section reads as follows: "*In our study, $I_C$ was calculated for each rain event and it is important to point out that the fraction of P lost via $I_C$ (i.e., $I_C/P$) during a single rain event varies in response to the magnitude and intensity of P (Gash, 1979;Linhoss and Siegert, 2016;Rutter et al., 1971;Zeng et al., 2000). The highest $I_C/P$ are expected to occur during light rainfall*

*events in a dry canopy, whereas I$_C$/P decreases with increasing rain amount and intensity as well as when water storage capacity in the canopy is reduced by intercepted water from previous precipitation events. Thus, projected changes in the amount and frequency of rainfall in northern latitude ecosystems (IPCC, 2014), could drastically alter I$_C$ and, in turn, strongly affect the amount of water available to plants, stream runoff and other downstream processes.*" (L546-554)

Figure S2 could be on your primary map.
We have removed Figure S2 and now include the location of the nodes where sap flow was measured in the map of the C2 subcatchment in Figure 1.

L 294 : solved ?
We have reworded this sentence. The sentence now reads as follows: "*We used measured soil moisture and soil temperature at the depth of 0.05 m as lower boundary conditions for the model.*" (L359-360)

L366 => T used a majority of available water.
We have rewrite this sentence to make it clear that transpiration was the largest water flux component during the study period. The sentence reads as follows: "*Canopy transpiration (T) was the largest ET flux component, and during 88% of the study period it alone was higher than Q (Fig. 3b).*" (L403-404)

L 374 => they include
We have rewritten this sentence. The sentence now reads as follows: "*Modeled estimates of intercepted P in the tree canopy together with understory evapotranspiration (I$_C$ + ETu) followed a similar pattern to the measured data, which here was computed as the difference between ET and T (Fig. 3c.)*" (L411-413)

L 376 => observed data ?
We have rewrite this sentence to make it clear that we are comparing modeled estimates to measured data. The sentence reads as follows: "*Modeled estimates of intercepted P in the tree canopy together with understory evapotranspiration (I$_C$ + ETu) followed a similar pattern to the measured data, which here was computed as the difference between ET and T (Fig. 3c).*" (L411-413)

L 385 => area represents or areas represent
We have change "areas represents" to "areas represent" The sentence now reads as follows; "*Colored shaded areas show simulation results for whole parameter space and gray shaded areas represent uncertainty in measurements.*" (L422-424)

figure 5 => incoming precipitation wasn't modeled, was it ? Understorey => Understory
Precipitation was not modelled. However, we understand how this could be misunderstood in the Figure 5. We now place precipitation in the middle of the figure with arrows going to both the "measured" (left side) and "modelled" (right side) partitioning approaches. Additionally, we now use the abbreviation of ET flux components in the figure as suggested by the reviewer #2.

L 438 - this is a long sentence
We have reworded this sentence. The sentence now reads as follows: "*Our findings clearly highlight the important role canopy trees play in the boreal hydrological cycle during the*

*growing season, and stresses the need to better understand the effect of trees and their response to forest management practices and a changing climate.*" (L486-489)

comparing => compare
We have reworded this sentence as suggested in the previous comment. The sentence now reads as follows: "*Our findings clearly highlight the important role canopy trees play in the boreal hydrological cycle during the growing season, and stresses the need to better understand the effect of trees and their response to forest management practices and a changing climate.*" (L486-489)

Maybe go one step further with a thought experiment. For example, based on our measurements, if all trees were removed from entire subcatchment, we would expect discharge over this period to go up by X%.
We agree that it is interesting to assess how changes in forest stand structure could influence the movement of water at our study site. Although we do not speculate how the removal of all trees would influence stream discharge, we do assess how changes in forest stand structure (*i.e.*, leaf area index; LAI) could influence ET and its flux components. In the discussion section, we have a paragraph where we discuss the potential consequences of how changes in forest stand structure, as a result of forest management practices, could affect the way water moves through boreal forests. The section reads as follows: "*Consequently, forest management practices that alter forest stand structure could have large cascading effects on the way water moves through these landscapes (Greiser et al., 2018). For instance, thinning reduces basal area and LAI of the remaining stand, whereas nitrogen fertilization in boreal forests promotes greater aboveground carbon allocation leading to an increase in LAI (Lim et al., 2015) and can also positively affect leaf photosynthetic efficiency and transpiration (Walker et al., 2014). To assess how forest management practices may affect ET as well as the relative importance of its component fluxes, we ran the APES model with canopy LAI values ranging from 1 to 7 $m^2 m^{-2}$. Over this LAI range, ET for the study period increased by ca. 50 mm (Fig. 6a). Fig. 6 also enabled us to identify thresholds in canopy LAI where the dominant ET component changes. For example, in sparse coniferous stands with LAI less than 3 $m^2 m^{-2}$, understory evapotranspiration appears as the dominant ET component flux, whereas in forest stands with LAI greater than 3 $m^2 m^{-2}$ transpiration becomes the dominant component (Fig. 6b). Understanding how LAI influences ET and its components fluxes provides an opportunity to assess how different forest management practices may affect the movement of water in forested landscapes. This, in turn, could assist in the development of more sustainable management practices (Stenberg et al., 2018;Sarkkola et al., 2013).*" (L587-604)

- this would have been interesting in introduction (with citations and further precision). In the introduction, could you have shown us an annual hydrograph and situation your study within that? (I know you showed a little bit of a hydrographic, but it wasn't the whole year). We changed figure 2 to include environmental data and the hydrograph for the entire 2016 year. We include dotted vertical lines in the figure to clearly show the study period used in this study.

Figure 6 - It is hard to believe these are lines.
We are a little confused about this comment. In Figure 6, we present the results of model simulations in which we ran the APES model where LAI was changed from 1 to 7 $m^2 m^{-2}$ at 0.5 intervals. In doing so, we show how changes in LAI influences ET and it flux components. We explain this in the discussion section where we say: "*To assess how forest*

*management practices may affect ET as well as the relative importance of its component fluxes, we ran the APES model with canopy LAI values ranging from 1 to 7 $m^2 m^{-2}$.* " (L593-595)

We also now place the two panels in Figure 6 next to each other to more clearly show how changes in LAI influences ET and its flux components.

**Reviewer #2: Dr. Miriam Coenders-Gerrits**

L146: the author only consider the growing season. I think it's important to emphasis throughout the paper.
We agree that this is an important point to emphasis and have made sure to highlight this throughout the manuscript. We have also made a change to the title to emphasis that this study only considers the growing season. The new title reads as follows: "*Partitioning growing season water balance within a forested boreal catchment using sapflux, eddy covariance and a process-based model*"

- L176: I am happy to see that below EC-system latent heat is considered. However, equation 1 is only valid once the forest is homogeneous since the footprint of the ECsystem and the below canopy latent heat are different. How 'homogeneous is your forest? Please elaborate. –
The C2 subcatchment is completely covered (99.9% forest cover) by an old growth (> 100 yr.) mixed forest stand. We now included a high-resolution aerial photograph of the C2 subcatchment in Figure 1c that shows the homogenous forest cover within the subcatchment. We have also added some text in the methods section to better describes the homogenous nature of the forest stand within the C2 subcatchment. The sentence reads as follows: "*The C2 subcatchment is completely covered by an old growth (>100 yr.) mixed forest stand of Picea abies (61 %), Pinus sylvestris (34 %), and Betula (5 %) (Laudon et al. 2013).*" (L149-150)

L199: the TF-sampling was done on 'event-base'. Please elaborate on how this was done. Did you run into your forest after rain ceased? Or did you do daily observations? How did you defined 'an event'?
We collected water from individual rain gauges immediately after rain ceased and thus each rainstorm represents an 'event'. During the study period that corresponded to 26 rain events. We added a sentence to the methods section that describes how the TF sampling was done. The added sentence reads as follows: "*Water was collected from individual rain gauges immediately after each rain event resulting in event-based $I_C$ estimates (Gash, 1979).*" (L227-228)

- L207: How did you tested whether ALS had the highest correlation with seasonal interception loss?
We used the FUSION software to characterized the canopy structure above each throughfall collector (*i.e*., a two-meter horizontal distance for each collector), which is based on spatial canopy density data acquired from airborne laser scanning (ALS). In total, the FUSION software gave us a total of 121 different matrices that describes the canopy structure. We then looked at the correlation coefficient between season IL for each of the 25 TF collectors and all 121 canopy matrices. We found that ElevMADmedian had the highest correlation with measured seasonal interception losses and could explain 77% of the variation in IL. We have added some text to the methods section to make this clear. We now also include a table in the supplementary material (Table S1) that shows the 10 canopy matrices that had the highest correlation with seasonal $I_C$, as suggested by reviewer #1. The section now reads as follows: "*Spatial canopy density data acquired from airborne laser scanning (ALS) was used in the FUSION software (McGaughey, 2012) to characterized the canopy structure above each throughfall collector (2 m radius around each collector). We found that the absolute deviation of ALS height measurements from overall median height (ElevMADmedium) showed the highest correlations to $I_C$ and could explain 77% of variation in seasonal $I_C$ (Table S1).*" (L228-233)

- L220-285: Please have a look at the recent technical note by Larsen et al 2019. Would it be necessary to compensate your sapflow measurements as well? Not doing this could mean an overestimation of your transpiration.

Thank you for bring this paper to our attention. The paper by Larsen et al. (2019) highlights the concerns of probe misalignment when using heat pulse sensors for sap flow measurements. In our study, we used the heat dissipation approach and it is unclear if probe misalignment has the same effect, or has any effect, and if it has an effect whether the proposed correction based on heat pulse sensors would work for heat dissipation sensors. Employing the correction therefore may increase the error.

In our study we accounted for known sources of sources associated with radial, azimuthal and trees size in an attempt to minimize errors association with our calculations of transpiration. Although we employed the same coefficients when calculating transpiration we believe this has a minimal effect because the approach we used has previously been shown to produce reasonable results, especially in conifers, based on comparisons with eddy covariance and mass balance approaches (Oren et al. 1998; Schäfer et al. 2002; Ward et al. 2008; Oishi et al 2008; Tor-ngern et al. 2018; Ward et al. 2018).

Oren R, Phillips N, Katul G, Ewers BE, Pataki DE (1998) Scaling xylem sap flux and soil water balance and calculating variance: a method for partitioning water flux in forests. *Annales des Sciences Forestieres* 55:191-216

Schäfer KVR, Oren R, Lai CT, Katul GG (2002) Hydrologic balance in an intact temperate forest ecosystem under ambient and elevated atmospheric $CO_2$ concentration. *Global Change biology* 8: 895-911

Ward EJ, Oren R, Sigurdsson BD, Jarvis PG, Linder S (2008) Fertilization effects on mean stomatal conductance are mediated through changes in the hydraulic attributes of mature Norway spruce trees. *Tree Physiology* 28: 579-596.

Oishi AC, Oren R, Stoy PC (2008) Estimating components of forest evapotranspiration: A footprint approach for scaling sap flux measurements. *Agricultural and Forest Meteorology* 148: 1719-1732

Tor-ngern P, Oren R, Palmroth S, Novick K, Oishi A, Linder S, Ottosson-Löfvenius M, Näsholm T (2018) Water balance of pine forests: synthesis of new and published results. *Agriculture and Forest Meteorology* 259:107-117

Ward EJ, Oren R, Kim HS, Kim D, Tor-ngern P, Ewers BE, McCarthy HR, Oishi AC, Pataki DE, Palmroth P, Phillips NG, Schäfer KVR (2018) Evapotranspiration and water yield of a pine-broadleaf forest are not altered by long-term atmospheric [$CO_2$] enrichment under native or enhanced soil fertility. *Global Change Biology* 24: 4841-4856. DOI: 10.1111/gcb.14363

- section 2.3: a better explanation of the modelling principles of APES, would help the reader. For example showing model-scheme.

We have reorganized and streamlined Section 2.3 to provide a better overview of the modeling principles of APES (L314-364). The reader can find a Figure of the model scheme in Launiainen et al. (2015), which we cite when describing the model.

Launiainen, S., Katul, G. G., Lauren, A., and Kolari, P. (2015) Coupling boreal forest $CO_2$, $H_2O$ and energy flows by a vertically structured forest canopy – Soil model with separate bryophyte layer, Ecological Modelling, 312, 385-405.

- section 2: I think it would help to make a schematic picture (a bit like figure 5) of how you define ET and its subcomponents.

We acknowledge that it is a little unclear on how exactly we define and quantify ET and its flux components. We therefore added a paragraph to the beginning of section 2 that explains how we calculated ET and it flux components. The paragraph reads as follows: "*We used the hydrological mass balance approach in combination with empirical measurements of vertical and horizontal water fluxes to quantify the water balance components within the C2 subcatchment. The mass balance equation is*

$$ds/dt = P - ET - Q \qquad (1)$$

*where ds/dt is change in soil water storage per unit area and Q is stream runoff. ET was measured using the eddy covariance technique, and partitioned into components as*

$$ET = T + I_C + ET_U \qquad (2)$$

*where canopy tree T was determined using sap flow sensors and evaporation of intercepted P from the tree canopy ($I_C$) was determined as the difference between open sky precipitation and water collected on event basis in rain gauges placed below the canopy (see below). Understory evapotranspiration (ETu) was not directly measured in this study, but was instead calculated as*

$$ET_U = ET - I_C - T \qquad (3)$$

*Because $I_C$ was estimated on an event basis, our estimate of ETu was for the entire growing season. Daily stream runoff (Q) was calculated as daily discharge, obtained from the Svartberget data portal (https://franklin.vfp.slu.se/), per catchment area. Change in soil water storage (ds/dt), which includes ground water recharge, was calculated as the residual of the hydrological mass balance (eq. 1).*" (L167-184)

We believe this added paragraph now provides a clear description of how we quantified ET and its flux components. However, we could also include a schematic picture if it is deemed necessary.

- L376-380: be careful with your definitions of transpiration, evaporation and evapotranspiration. ETu is a combination of forest floor interception, understory transpiration (mosses) and soil evaporation and is thus not only 'evaporation' as said in L378. Also the role of soil evaporation is not explained. Is soil evaporation relevant in your study site? Why/why not.

We have carefully gone through the manuscript to make sure we are consistent with our definitions of transpiration, evaporation, and evapotranspiration. Additionally, we have change "*IL*" to "*Ic*" throughout the manuscript to make it clear that we are talking about evaporation of intercepted precipitation in the tree canopy. In this specific case (L376-380), we have rewritten this sentence to make it clear that we are talking about *Ic* and understory evapotranspiration (*ETu*). The sentence reads as follows: "*Modeled estimates of intercepted P*

*in the tree canopy together with understory evapotranspiration (I$_C$ + ETu) followed a similar pattern to the measured data, which here was computed as the difference between ET and T (Fig. 3c).*" (L411-413)

At our site, soil evaporation is negligible as there is very little bare ground within the C2 subcatchment. We now provide this information in the methods section when describing the understory vegetation at our site. The sentence reads as follows: "*The understory consists of a continuous layer of bilberry (Vaccinium myrtillus), lingonberry (Vaccinium vitis idaea), and mosses (Pleurozium schreberi and Hylocomium splendens) with no bare ground.*" (L151-153)

- Section 4: the discussion and conclusions are merged into one section. I think it would be better to split this. And/or merge the discussion with the results section. But for sure make a separate section for the conclusions only where you are only answering to the research objective.
We have now made a separate section for the conclusions.

Specific (minor) comments:
- L31: redundant to mention "and being roughly 7 times greater than stream runoff". This is the same info as saying ET is 85
We have removed "and being roughly 7 times greater than stream runoff" from the sentences. The sentence now reads as follows: "*During the growing season, ET represented ca. 85 % of the incoming precipitation.*" (L29-30)

- L44: Maybe better to mention the spread in global ET. This is ca 55-80
We now mentioned the spread in global ET. The sentence now reads as follows: "*At a global scale, ET accounts for ca. 60 % of the annual terrestrial P (Oki and Kanae, 2006); yet the relative importance of ET varies considerably among different biomes, ranging between 55–80 % of incoming P (Peel et al., 2010).*" (L43-47)

- L71: after e.g. a comma.
We added a comma after e.g. The sentence now reads as follows: "*Most studies typically partition ET at the stand or plot scale without considering the broader hydrological cycle (e.g., Cienciala et al., 1997;Grelle et al., 1997;Wang et al., 2017;Ohta et al., 2001;Iida et al., 2009;Hamada et al., 2004;Maximov et al., 2008;Warren et al., 2018;Schlesinger and Jasechko, 2014).*" (L89-92)

- L128: unit of annual rainfall is mm/year.
We now include yr-*1* in our units of annual rainfall. The sentence now reads as follows: "*Mean annual precipitation is 619 mm yr$^{-1}$, with the majority (ca. 60%) falling in the form of rain.*" (L132-133)

- L157-165: variables like P, Q, dS, etc should be in italic.
We now italicized all water flux components (i.e., *P, Q, ET, T, I$_C$, ETu*, and *ds/dt*) in this section and throughout the manuscript.

- L165: I prefer to rename dS into dS/dt, since dS is the storage change per time.
We have changed ΔS to *ds/dt* in this section as well as throughout the manuscript.

- L172: details => detail.

We have rewritten this sentence as suggested by reviewer #3. The sentence now reads as follows: "*A detailed description of the EC data processing and quality control can be found in Chi et al. (2019).*" (L200-201)

- Fig S1: the unit of P is mm/y. Furthermore, I would change instead of showing Q/P, showing ET/P. Since this the focus of the paper.
We changed the units of P to mm yr$^{-1}$ and now show ET/P in Figure S1.

- L337-342: This is a result.
We agree that L337-342 can be interpreted as a result, but we consider this finding as a test of the validity of the model at our study site. As the APES model was able to represent individual components of the surface energy balance reasonably well, it gives us confidence on the model's predictions of *ET* and its flux components. This information is only used as a model check and thus we choose to present it in this section and as a supplementary figure.

- Fig3c: why showing IL+ETu? Why not only ETu? This would more sense in my view.
We agree that it would be nice to directly compared daily values of "measured" and "modeled" ETu during the study. However, this was not possible because canopy interception loss (IL) were determined on an event-basis, and not on a daily basis. The "measured" data presented in Figure 3c is the difference between ET and canopy transpiration, which is $I_C$ + ETu. We have rewritten the figure caption to make this clearer. The figure caption now reads as follows: "***Figure 3**. Measured and modelled evapotranspiration ET (a) and its component fluxes: canopy transpiration, T (b), evaporation of intercepted P in the tree canopy and understory evapotranspiration, $I_C$ + ETu (c) and modeled canopy interception evaporation, $I_C$ (d) in a boreal forest catchment during the 2016 growing season. Colored shaded areas show simulation results for whole parameter space and gray shaded areas represent uncertainty in measurements. Small panels on the left side show correlation between daily modelled and measured values. Measured $I_C$ + ETu in panel (c) was determined as the difference between total ET and T.*" (L419-426)

- Section 2/fig 3: explain how ETu is 'measured'. It's calculated as ETu=ET-IL-T, right? Please add this equation and elaborate on the fact that ETu is thus not independent of the other measured components.

Understory evapotranspiration (*ETu*) was not directly measured in this study, but instead was calculated as: *ETu = ET – $I_C$ – T*. Moreover, because $I_C$ was estimated on an event basis, our estimate of *ETu* was for the entire growing season. We added text to the methods section that better describes how *ETu* was calculated. The section reads as follows: "*Understory evapotranspiration (ETu) was not directly measured in this study, but was instead calculated as*

$$ET_U = ET - I_C - T \qquad\qquad (3)$$

*Because $I_C$ was estimated on an event basis, our estimate of ETu was for the entire growing season.*" (L177-181)

- Figure 5: I would add the percentages as well. Furthermore, be consistent in the naming of ET and its subcomponents. Would it not be better to use here the abbreviations?
In Figure 5, we now include the percentage of individual flux in relation to total ET. We did not include the percentage of individual flux components in relation to incoming P, as we believe this may cause confusion and would make the figure more difficult to understand. However, the values of total P and individual water pathways are presented in this figure, which makes it possible to also determine the percentage of different water pathways in relation to total P. The caption to Figure 5 now reads as follows: "***Figure 5***. *Partitioning of water fluxes based on empirical measurements (left side) and model simulation (right side) in a coniferous boreal catchment during the 2016 growing season (July-October). Values for each flux are presented as mean absolute values (mm) with upper and lower boundaries shown in parenthesis. The percentages gives the relative contribution of ET components to total ET.*" (L457-461)

Additionally, we now use the abbreviation for the different ET flux components in the figure.

**Reviewer #3: Anonymous reviewer**

Title ăĂ˘ c The word composition of the title is not clear " ´ . . .forest water balance. . ." is it partitioning of water balance in boreal forest, or partitioning forest-water balance?
We have changed the title to: "*Partitioning growing season water balance within a forested boreal catchment using sapflux, eddy covariance and a process-based model*"

Abstract:
ăĂ˘ c It would be nice to see water balance ways more specific to boreal ´ forests to get a clearer picture how this work is worthy for readers
In the abstract, we now make it clear that few studies have partitioned ET into it individual flux components in boreal forests. The first sentence of the abstract now reads as follows: "*Although it is well known that evapotranspiration (ET) represent an important water flux at local to global scales, few studies have quantified the magnitude and relative importance of ET and its individual flux components in high latitude forests.*" (L21-23)

Also, in the introduction we now highlight the considerable variation in the relative importance of ET in boreal forests. Thus, quantifying the magnitude and spatiotemporal variation of transpiration and evaporation separately is crucial to better understand how water moves through boreal forest landscapes.

ăĂ˘ c In line 20, it ´ reads "water is lost"; this is very confusing wording all over the paper. 1) water cannot be lost from a system, 2) I assume this paper deals with water balance, so water "flows" from one state/regime to next, and that is not lost, 3) there could be some cases where ET can be referred as lost; that is when rainfall is dealt as "gain"
We agree that ET is a water flux and that it may be misleading, and potentially confusing, to consider ET as a "loss". We have carefully gone through the manuscript and replaced "loss" with ET and its component fluxes and no longer refer to ET as a "water loss". We have also rephrased the first sentence in the introduction to now describe the movement of water in terrestrial ecosystems as inputs and outputs as suggested by reviewer #1. The sentence now reads as follows: "*In the hydrological cycle, water enters terrestrial ecosystems mainly through precipitation (P). This water leaves terrestrial ecosystems either through evapotranspiration (ET) back to the atmosphere or as stream runoff (Q).*" (L41-43)

ăĂ˘ c Line ´ 30 change "water loss pathway" to "water balance component"
We have changed "water loss pathway" to "water balance components". The sentence now reads: "*This study was conducted within the Krycklan Catchment, which has a rich history of hydrological measurements thereby providing us the unique opportunity to compare the*

*absolute and relative magnitude of ET and its flux components to other water balance components.*" (L26-29)

âAˇ c Line 32 Canopy ´ interception is not part of ET, it should be rather evaporation from canopy
We agree that interception in not part of ET, but rather evaporation of intercepted water in canopy trees. We have rewritten this sentence to make this clear. The sentence now reads as follows: "*Both empirical results and model estimates suggested that tree transpiration (T) and evaporation of intercepted water from the tree canopy ($I_C$) represented 43 % and 31 % of ET; respectively, and together was equal to ca. 70 % of incoming precipitation during the growing season.*" (L29-33)

âAˇ c Line 33- ´34, the numbers do not add up 70, check
We agree that the numbers in line 33-34 do not add up to 70. However, the number presented in lines 33-34 represented the percentage of T and $I_C$ to total ET, whereas the 70 % is in reference to T and IL being equal to ca. 70 % on the incoming precipitation during the growing season. We have reworded this sentence to make this clear. The sentence now reads as follows: ""*Both empirical results and model estimates suggested that tree transpiration (T) and evaporation of intercepted water from the tree canopy ($I_C$) represented 43 % and 31 % of ET; respectively, and together was equal to ca. 70 % of incoming precipitation during the growing season.*" (L29-33)

Introduction:
âAˇ c The study has got no clear ´ definition of hypothesis or purpose of the study
The objectives of the study are stated in the final paragraph of the introduction: The main objective of this study was to *i*) constrain the absolute and relative magnitudes of *ET* flux components by using both empirical data and model simulations, *ii*) to explore how they vary during the course of the growing season, *iii*) to compare different ET flux components to other water balance components *(i.e.*, stream runoff) and *iv*) directly assess the important role trees play in the boreal hydrological cycle during the growing season.

âAˇ c Line 51-52, I don't agree that most ´ studies treat ET as a single water flux pathway
We have removed this sentence from the manuscript.

âAˇ c Line 62-63, I think, rather there ´ are dozens of experimental studies for decades
We have reword this sentence and now acknowledge the long history of research on ET as suggested by reviewer #1. The sentence now reads as follows: "*Research investigating the biotic and abiotic controls on ET has a long history, dating back centuries (Katul et al., 2012;Brutsaert, 1982). However, efforts to separately estimate T and evaporation began in the 1970s (see Kool et al., 2014) and ever since there has been an increasing number of studies partitioning ET (Stoy et al., 2019;Schlesinger and Jasechko, 2014). There are a number of different approaches and methodology to partition ET into its individual flux components (Kool et al., 2014), including empirical measurements (Mitchell et al., 2009;Cavanaugh et al., 2011;Good et al., 2014;Sutanto et al., 2014) as well as a number of different process based models (Sutanto et al., 2012;Stoy et al., 2019;Launiainen et al., 2015).*" (L72-80)

âAˇ c Line 73, what does it mean by ´ "few investigation on water balance at catchment scale"?

We are trying to highlight that the majority of ET partitioning studies have been done at the stand and/or plot scale and thus are not able to directly compare the magnitude of ET and its flux components to other water pathways (i.e., steam runoff). We have rewritten this sentence to make this clearer. The sentence now reads as follows: "*We are aware of only a few investigations that have partitioned ET at the catchment scale (Telmer and Veizer, 2000;Sarkkola et al., 2013), and thus we have little empirical data about how T compares to other water fluxes (i.e., streamflow) in the terrestrial hydrological cycle.*" (L92-97)

âA˘ c The paragraph after line ´ 90 better fits above the previous paragraph
We agree and have moved this section to the previous paragraph.

âA˘ c Line 114, what is the state-of-the-art ´ of hydrological measurements at the study site? Give some details of measurements done which of course respective to this study
We have rewritten this sentence to highlight that this study builds upon the rich history of long-term hydrological measurements within the Krycklan catchment. The sentence now reads as follows: "*This study was conducted within the Krycklan Catchment, which has a rich history of hydrological measurements (see Laudon et al., 2013;Laudon and Sponseller, 2018), thereby providing us the unique opportunity to compare different ET flux components to other water balance components (i.e., streamflow) as well as to directly assess the important role trees play in the boreal hydrological cycle.*" (L118-122)

Methods:
âA˘ c Line 147-148, not clear ´
We have removed "spanning from after the spring flood until leaf senescence for deciduous species" from the sentence. The sentence now reads as follows: "*Our study period was the growing season of 2016. The water balance and ET partitioning were restricted to July-October due to measurement availability.*" (L159-161)

âA˘ c Line 153-155, not clear
We have removed this sentence from the manuscript.

â A˘ c Line 157, what are the environmental data, give the ´ details or examples
We now provide details about the instruments used to measure environmental data. The sentences now reads as follows: "*Environmental data used in this study included open sky precipitation (T200BM Geonor Inc., New Jersey, USA), air temperature and relative humidity (MP102H Rontronic AG, Switzerland), wind speed (METEK uSonic3 Class-A, Meteorologische Messtechnik GmbH, Germany), atmospheric pressure (PTB210 Vaisala Inc., Finland), incoming short and long-wave radiation (CNR4 Kipp & Zonen B.V., Netherlands), photosynthetic active radiation (PAR; SQ-110 Apogee Instruments Inc., Utah, USA), as well as soil temperature and moisture measured at 0.05 m depth (Thermocouple, Type E Campbell Scientific Inc., Utah, USA). All environmental data were obtained from the ICOS portal, Svartberget station (http://www.icos-sweden.se/data.html).*" (L185-193)

âA˘ c Paragraph line 165-175, Too much information. Please classify ´ with instruments, data, how processed, calibrated, purpose – this might help readers to understand
We have simplified and clarified the description of the eddy covariance measurements in the methods section as suggested. The section now reads as follows: "*The EC instrumentation consists of a 3D ultrasonic anemometer (METEK uSonic3 Class-A, Meteorologische Messtechnik GmbH, Germany) for measuring wind components (u, v, w) and an enclosed infrared gas analyzer (LI-7200, LI-COR Biosciences, USA) for measuring $CO_2$ and $H_2O$*

*concentrations. The 10 Hz raw data were processed in the EddyPro® software (version 6.2.0, LI-COR Biosciences, USA) to obtain the 30-min averaged fluxes. A detailed description of the EC data processing and quality control can be found in Chi et al. (2019). In brief, the half-hourly ET data were corrected for changes in the storage term which was estimated from concentration profile measurements at several levels (4, 10, 15, 20, 25 and 30 m) between the forest ground and the measurement height. ET data were then filtered based on the EddyPro quality check flagging policy which includes tests on steady state and developed turbulent conditions based on Mauder and Foken (2004), advection effects (Wharton et al., 2009), wind distortion, power failure, and site maintenance activities. Gaps in the half-hourly ET data were filled based on empirical relationships between ET and net radiation using the REddyProcWeb online tool (Wutzler et al., 2018). Based on the Kljun footprint model (Kljun et al., 2015), the EC footprint (90 %) covers a measurement area of ~0.5 km² with a mean upwind fetch of ~400 m surrounding the tower. The uncertainty in the EC-based ET was estimated by the Monte Carlo simulation (Richardson and Hollinger, 2007)."* (L195-212)

 âAˇ c Line 179, what does it mean by "non-stationarity" this word commonly ´ used in statistical description not in instrumentation
This sentence has been rephrased and now reads as follows: "*ET data were then filtered based on the EddyPro quality check flagging policy which includes tests on steady state and developed turbulent conditions based on Mauder and Foken (2004), advection effects (Wharton et al., 2009), wind distortion, power failure, and site maintenance activities.*" (L203-207)

âAˇ c Assumptions described in line ´ 188-190 are wrong, re-write (it should be IL = GP-TF-SF)
We are aware that stemflow (SF) is often included when calculating canopy interception losses (i.e., $I_C = GP - TF - SF$). However, previous work within the Krycklan catchment has shown no SF in forest stands dominated by spruce and pine trees during the summer months (Venzke 1990). Thus, we have omitted SF when calculating $I_C$ in our study. We have added a sentence in the methods sections that highlights this previous observation which in turn provides justification for our calculation of $I_C$ as the difference between GP and TF. The section now reads as follows: "*Evaporation of intercepted P from the tree canopy ($I_C$) was determined by subtracting throughfall (TF) from open sky P:*
$$I_C = P - TF \qquad\qquad\qquad (4)$$
*Previous research within the Krycklan catchment has shown that during the growing season stemflow is negligible in forest stands dominated by P. sylvestris and P. abies (Venzke, 1990) and consequently omitted in this study.*" (L213-218)

Results and discussion
âAˇ c´ Are mixed up and not well structured: please take rendering sentences from results to discussion
We have carefully gone through the results and discussion sections to better improve its structure as well as make sure that all interpretation of the data is moved to the discussion section.

[revised manuscript text omitted]